# SynRS3D: A Synthetic Dataset for Global 3D Semantic Understanding from Monocular Remote Sensing Imagery

**Jian Song**[1,2], **Hongruixuan Chen**[1], **Weihao Xuan**[1,2], **Junshi Xia**[2], **Naoto Yokoya**[1,2]

[1]The University of Tokyo, Tokyo, Japan
[2]RIKEN AIP, Tokyo, Japan
`song@ms.k.u-tokyo.ac.jp`
 https://JTRNEO.github.io/SynRS3D

## Abstract

Global semantic 3D understanding from single-view high-resolution remote sensing (RS) imagery is crucial for Earth observation (EO). However, this task faces significant challenges due to the high costs of annotations and data collection, as well as geographically restricted data availability. To address these challenges, synthetic data offer a promising solution by being unrestricted and automatically annotatable, thus enabling the provision of large and diverse datasets. We develop a specialized synthetic data generation pipeline for EO and introduce *SynRS3D*, the largest synthetic RS dataset. SynRS3D comprises 69,667 high-resolution optical images that cover six different city styles worldwide and feature eight land cover types, precise height information, and building change masks. To further enhance its utility, we develop a novel multi-task unsupervised domain adaptation (UDA) method, *RS3DAda*, coupled with our synthetic dataset, which facilitates the RS-specific transition from synthetic to real scenarios for land cover mapping and height estimation tasks, ultimately enabling global monocular 3D semantic understanding based on synthetic data. Extensive experiments on various real-world datasets demonstrate the adaptability and effectiveness of our synthetic dataset and the proposed RS3DAda method. SynRS3D and related codes are available at `https://github.com/JTRNEO/SynRS3D`.

## 1 Introduction

3D reconstruction is a fundamental task in computer vision, which focuses on creating three-dimensional representations from two-dimensional images. It plays a crucial role in applications such as virtual reality, autonomous driving, and robotics. In the context of Earth observation (EO), reconstructing semantic 3D information from single-view remote sensing (RS) images is also vital for applications like environmental monitoring, urban planning, and disaster response [50, 52, 53, 65]. Unlike point cloud-based 3D reconstruction, which relies on LiDAR or stereo cameras, monocular semantic 3D reconstruction is more scalable and requires less expensive equipment, making it suitable for global applications [21, 87, 120, 44, 50, 52, 53, 65, 54, 26, 22, 49]. This task combines land cover mapping, which is closely related to semantic segmentation in computer vision, and height estimation. However, acquiring RS annotations is costly and time-consuming, especially for high-resolution height data obtained through satellite LiDAR (e.g. GEDI, ICESat-2) [85, 30, 51] or stereo matching [3, 116, 112, 57, 27, 64]. Furthermore, high-resolution land cover mapping datasets [16, 96, 101]

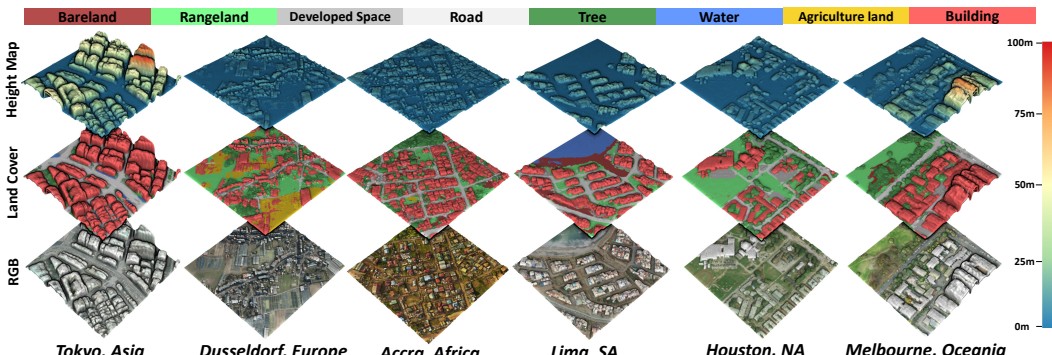

Figure 1: 3D visualization outcomes from real-world monocular RS images, using the model trained on the SynRS3D dataset with the proposed RS3DAda method. The top colorbar represents the legend for the land cover map (row 2), while the right colorbar indicates the height range (row 1). "SA" indicates South America and "NA" indicates North America.

often lack the corresponding height data. Moreover, the availability of RS data is geographically skewed, with developed regions having abundant data and developing regions lacking high-resolution datasets. Schmitt et al. [83] reviewed more than 380 RS datasets, revealing that few datasets come from Oceania, South America, Africa and Asia, while most originate from Europe and North America. This geographic limitation in RS datasets raises a crucial question: Can findings from numerous research papers be applied to these underrepresented regions?

The aforementioned challenges can be effectively addressed using synthetic data. Current 3D modeling technology has the potential to create various landscape features with accompanying land cover semantic labels and height values. Therefore, we present *SynRS3D*, a high-quality, high-resolution synthetic RS 3D dataset. SynRS3D includes 69,667 images with various ground sampling distances (GSD) ranging from 0.05 to 1 meter, and annotations for height estimation and land cover mapping in various scenes. However, models trained solely on synthetic data tend to overfit to these datasets, resulting in significantly reduced performance when applied to real-world environments due to the large domain gap. Existing synthetic datasets [7, 124, 43, 84, 105, 75, 76, 86, 104] often exhibit this significant performance gap compared to models trained on real data.

To bridge this gap, we introduce *RS3DAda*, a novel baseline aimed at advancing research on SynRS3D and setting a benchmark for multi-task unsupervised domain adaptation (UDA) from synthetic to real scenarios of RS. This approach utilizes a self-training framework and incorporates a land cover branch to enhance the quality of pseudo-labels of the height estimation branch, thus stabilizing RS3DAda training and boosting the accuracy of both branches. For the height estimation task, our model even outperforms models trained on real-world data in the challenging areas. Figure 1 shows the results of the 3D semantic reconstruction globally using models trained with our RS3DAda method on the SynRS3D dataset. Furthermore, we include disaster mapping results for earthquake and wildfire scenarios using our model in Appendix A.9, showcasing its efficacy in real-world disaster response applications.

The major contributions of this work can be summarized in three aspects:

- We propose **SynRS3D**, the largest RS synthetic dataset with comprehensive annotations and geographic diversity for remote sensing tasks.

- We design **RS3DAda**, a robust and effective multi-task UDA algorithm for land cover mapping and height estimation.

- Based on SynRS3D, we benchmark various remote sensing scenarios for land cover mapping and height estimation tasks.

We hope that our dataset and the proposed method advance the progress of synthetic learning in remote sensing applications.

## 2 Related Work

**High-Resolution Earth Observation.**   High-resolution RS technology enables us to capture images with a GSD of less than 1 meter, significantly enhancing our understanding of Earth's surface details. Deep learning has become a powerful tool for analyzing these images. High-resolution imagery allows for precise land cover mapping [55, 121, 34, 97, 60, 102]. Concurrently, height estimation research [54, 26, 22, 49] focuses on determining accurate surface elevations. Some studies [21, 87, 120, 44] combine these tasks, training models for both land cover mapping and height estimation simultaneously. Most 3D reconstruction studies use real-world data, concentrating on buildings and often neglecting valuable classes like trees [50, 52, 53, 65]. Multi-view RS for 3D reconstruction [81, 20, 38, 19, 113, 47] is expensive and geographically limited. Benchmark datasets [16, 96, 101, 106, 45, 73, 13, 80] have been constructed for model training; however, acquiring high-resolution RS data is costly and time-consuming due to manual labeling and sophisticated equipment requirements. This limits the number of samples available to train robust models. Additionally, real datasets often lack geographic diversity, which can hinder model generalization to new, unseen areas.

**Synthetic Remote Sensing Dataset.**   Modern methods to create synthetic data utilize deep learning generative models, such as diffusion models [31] and generative adversarial networks (GANs) [25], in conjunction with 3D modeling techniques. Generative models often struggle to produce data outside their training distribution and lack the precise control needed for RS tasks. In contrast, 3D modeling approaches in computer graphics, which take advantage of game engines or 3D software [8, 66, 79, 77, 28, 100], have shown more success. However, creating synthetic data for RS is inherently more complex. A single 1024x1024 RS image demands hundreds of buildings, thousands of trees, and various topological features such as rivers and roads, unlike street views that require fewer assets. Recent studies have used 3D software to synthesize RS data with automatic annotations [7, 124, 43, 84, 105, 75, 76, 86, 104]. Despite their utility, these datasets often suffer from limited geographic diversity [7, 124, 105, 75, 76, 104], semantic categories [7, 124, 43, 84, 105, 75, 76, 104], and comprehensive semantic and height information [7, 124, 43, 84, 105, 86], which impacts their effectiveness in training robust, globally applicable models.

**Unsupervised Domain Adaptation (UDA).**   UDA aims to adapt a model trained on labeled data from a source domain to perform well in an unlabeled target domain, reducing the constraints and costs of data annotation. For semantic segmentation, most of the work focuses on adversarial learning [92, 93, 33, 32, 23, 98, 95, 91, 63] and self-training [122, 56, 115, 123, 67, 82, 110, 94, 35, 36, 37, 48, 24]. Unlike semantic segmentation tasks, height estimation is a regression task, aligning closely with monocular depth estimation. For UDA in monocular depth estimation, methods often focus on image translation to reduce domain gaps [4, 117, 119, 12], and some use self-training with pseudo-labels [61, 107, 111]. In RS, most research [40, 62, 72, 89, 90, 118] focuses on applying developed techniques from computer vision to adapt models trained on real-world data to different real-world environments (real-to-real). However, only a small number of studies [58, 103] investigate the challenges of adapting synthetic data to real-world environments (synthetic-to-real). To the best of our knowledge, RS3DAda is the first work to explore UDA algorithms specifically designed for synthetic-to-real domain adaptation for multi-task dense prediction in RS.

## 3 The SynRS3D Acquisition Protocol and Statistical Analysis

Although simulating RS scenes poses significant challenges due to the need for numerous assets, we mitigate this issue using procedural modeling techniques [69, 68, 42]. Instead of manually modeling each element, we incorporate rules derived from real-world knowledge, formalized into scripts. The generation system is controlled through hyperparameters like city style, asset ratio, and texture style, allowing us to create a diverse high-quality synthetic dataset. Table 1 compares existing synthetic RS datasets with SynRS3D, highlighting our dataset's advantages in diversity, functionality, and scale.

### 3.1 Statistics for SynRS3D

The left section of Figure 2 shows RGB images, land cover labels, and height maps for six styles in SynRS3D, representing diverse real-world environments. The bottom left section compares the height distribution of SynRS3D with two leading synthetic datasets, SMARS [76] and GTAH [104],

Table 1: Comparisons of various RS synthetic datasets based on diversity, image capture details, asset origins, tasks, and the number of images. The diversity is categorized into City-Replica (datasets mimicking specific cities) and Style-Extended (covering a range of urban styles). Image capture attributes include GSD, resolution, and perspective (Nadir, Oblique). Asset origins are denoted as Manually-made (M), Game-origin (G), Procedurally-generated (P), and Real (R). Tasks cover Change Detection (CD), Building Segmentation (BS), Object Detection (OD), Disparity Estimation (DE), Height Estimation (HE), Land Cover (LC), and Building Change Detection (BCD).

| RS Synthetic Datasets | Diversity | | Image Capture | | | Assets | | | Task | # Images |
|---|---|---|---|---|---|---|---|---|---|---|
| | City-Replica | Style-Extended | GSD (m) | Image Size | Perspective | Layout | Geometry | Texture | | |
| AICD [7] | ✓ | ✗ | * | $800 \times 600$ | Nad., Obl. | M | M | M | CD | $\sim 1K$ |
| GTA-V-SID [124] | ✓ | ✗ | 1.0 | $500 \times 500$ | Nad. | G | G | G | BS | $\sim 0.12K$ |
| Synthinel-1 [43] | ✓ | ✓ | 0.3 | $572 \times 572$ | Nad. | R | M | M+R | BS | $\sim 1K$ |
| RarePlanes [84] | ✓ | ✗ | $0.31 - 0.39$ | $512 \times 512$ | Nad., Obl. | R | M | M+R | OD | $\sim 65K$ |
| SyntCities [75] | ✓ | ✗ | 0.1, 0.3, 1.0 | $1024 \times 1024$ | Nad. | R | M | M+R | DE | $\sim 8K$ |
| GTAH [104] | ✓ | ✗ | * | $1920 \times 1080$ | Nad., Obl. | G | G | G | HE | $\sim 28.6K$ |
| SyntheWorld [86] | ✗ | ✓ | $0.3 - 1.0$ | Various | Nad., Obl. | P | P+M | P | LC, BCD | $\sim 40K$ |
| SMARS [76] | ✓ | ✗ | 0.3, 0.5 | Various | Nad. | R | M | M+R | LC, HE, BCD | 4 |
| SynRS3D (Ours) | ✗ | ✓ | $0.05 - 1.0$ | $512 \times 512$ | Nad., Obl. | P | P+M | P | LC, HE, BCD | $\sim 70K$ |

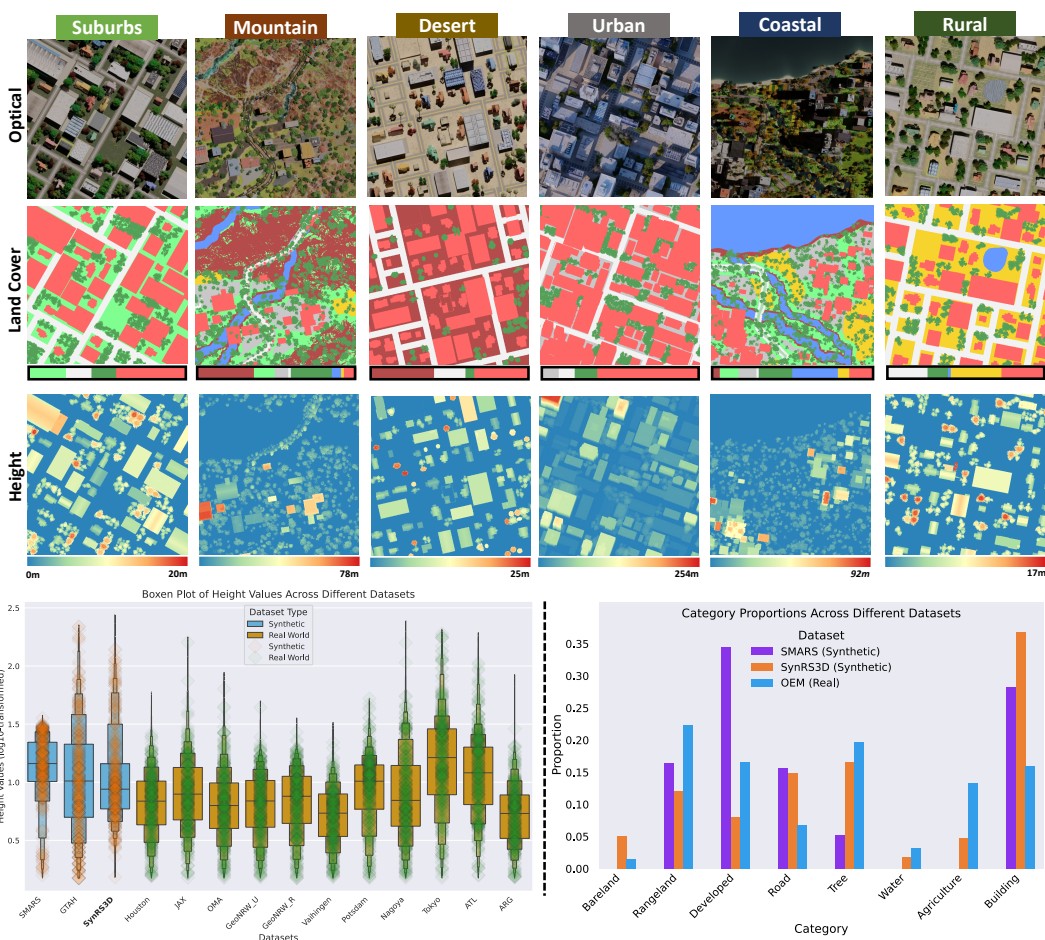

Figure 2: Examples and statistics of SynRS3D. The colorbar corresponds to the land cover classification legend shown in Figure 1.

as well as 11 real-world height datasets. SynRS3D's height distribution closely matches real-world data, while SMARS and GTAH show limitations. Specifically, SMARS, which mimics Paris and Venice, has a narrow height range. GTAH, based on the GTAV game, which mimics Los Angeles, shows a wider height range, but with a larger mean and variance, making it less representative of other cities. SynRS3D was constructed using the following prior knowledge [99]: backward regions (low buildings) cover about 12% of the world's areas, emerging regions (mid buildings) cover about

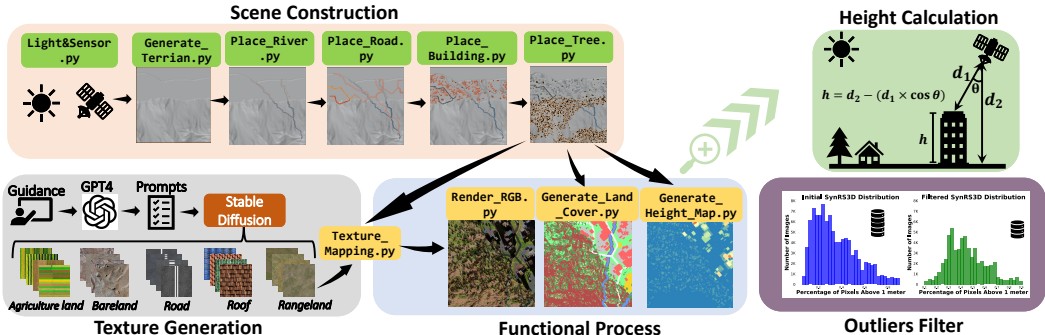

Figure 3: Generation workflow of SynRS3D.

70%, and developed regions (tall buildings) cover about 18%. The bottom right section contrasts the land cover proportions in SynRS3D with those in SMARS and the real-world OEM dataset [101], which is currently the largest and most geographically diverse dataset. SynRS3D's land cover categories—including Bareland, Rangeland, Developed Space, Roads, Trees, Water, Agricultural Land, and Buildings—match well with the OEM dataset. In contrast, SMARS has only five categories, limiting its effectiveness for comprehensive land cover mapping.

## 3.2   Generation Workflow of SynRS3D

The generation process of SynRS3D employs tools such as Blender [14], Python, GPT-4 [2], and Stable Diffusion [78], as illustrated in Figure 3. It begins with Python scripts that translate synthetic scene rules into parameter-controlled instructions for tasks like terrain generation, sensor placement, and asset placement. The geometry of the buildings and trees is created both procedurally and manually. Stable Diffusion generates textures based on detailed text prompts from GPT-4, ensuring high-quality and diverse textures. Blender's compositor node and Python scripts then generate accurate land cover labels and height maps. The details of the height map generation are detailed in the height calculation section of Figure 3. Our dataset's height maps are produced within Blender using simple geometric algorithms, resulting in a completely accurate normalized Digital Surface Model (nDSM). An nDSM represents the height of objects above the ground, providing clear information about buildings and vegetation. In contrast, the height maps for the comparative datasets GTAH and SMARS are Digital Surface Models (DSM), which include the height of ground and objects. Converting DSM to nDSM requires additional processing using the dsm2dtm [1] algorithm, which introduces noises. Optical images are produced using rendering scripts. To generate building change detection masks, we follow a structured process. Initially, buildings are randomly removed from scenes, and textures are reapplied to create pre-event images. Subsequently, land cover labels are subtracted to produce the change detection masks. After the initial generation of the dataset, images with anomalous height distributions are filtered out. This step ensures that the final version of SynRS3D closely aligns with real-world height distributions. The specific filtering algorithm used and examples of building change detection masks can be found in the Appendix A.1.

## 4   Multi-Task Unsupervised Domain Adaptation for RS Tasks (RS3DAda)

SynRS3D features low costs, high diversity, and large volume. However, there is a clear domain gap between synthetic data and real-world environments, which limits the use of SynRS3D. This limitation is particularly evident in RS, where synthetic-to-real UDA algorithms are lacking. To bridge this gap, we developed RS3DAda, which leverages land cover labels and height values to complement each other. In addition, it harnesses the potential of unlabeled real-world data, establishing the first benchmark for synthetic-to-real RS-specific multi-task UDA.

---

[1]https://github.com/seedlit/dsm2dtm

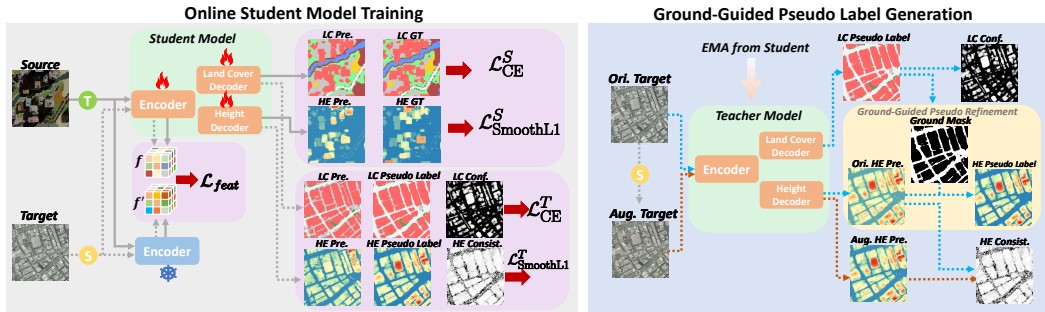

Figure 4: Overview of the proposed RS3DAda method. T denotes statistical image translation, S represents strong augmentation. For Online Student Model Training, dotted line: target image, solid line: source image. For Ground-Guided Pesudo Label Generation, dotted line: original target image, dotted line: strong augmented target image.

**Basic Framework.** In this work, we adopt self-training [46] as our basic UDA technique due to its superior stability and adaptability across both land cover mapping and height estimation tasks. Additionally, the teacher-student framework [88] is incorporated into the method to enhance performance and generalizability by stabilizing training with the exponential moving average (EMA) of model weights. The synthetic dataset image set $\mathcal{X}_s$ serves as the source domain, with access to corresponding land cover labels $\mathcal{Y}^s_{LC}$ and height maps $\mathcal{Y}^s_H$. The real dataset image set $\mathcal{X}_t$ serves as the target domain without any access to the labels.

**Source Domain Training.** In this stage, shown in the left part of Figure 4, source domain images $\mathcal{X}_s$ undergo statistical image translation using simple Fourier Domain Adaptation (FDA) [110], Histogram Matching (HM) and Pixel Distribution Matching (PDA), which follow the conclusion of Abramov et al. [1]. This process aligns the styles of the source images with the target domain, resulting in translated images $\mathcal{X}'_s$. These translated images from the source domain are then fed into the student network, producing predicted land cover labels $\hat{Y}^s_{LC}$ and height $\hat{Y}^s_H$. The supervised loss for the source domain is defined as:

$$\mathcal{L}_{source} = \frac{1}{N} \sum_{i=1}^{N} \left( \mathcal{L}_{CE}(\hat{Y}^{s(i)}_{LC}, \mathcal{Y}^{s(i)}_{LC}) + \mathcal{L}_{SmoothL1}(\hat{Y}^{s(i)}_H, \mathcal{Y}^{s(i)}_H) \right), \tag{1}$$

where $\mathcal{L}_{CE}$ is the cross-entropy loss, $\mathcal{L}_{SmoothL1}$ is the Smooth L1 loss [39], and $N$ is the total number of samples.

**Land Cover Pseudo-Label Generation.** To generate high-quality pseudo-labels for land cover mapping, as illustrated in the right section of Figure 4, target domain images $\mathcal{X}_t$ are strongly augmented denoted as $\mathcal{X}'_t$. We adopt color jitter, Gaussian blur, and ClassMix [70] as the strong augmentations. The teacher model then predicts the land cover pseudo-labels $\tilde{Y}^t_{LC}$. After that, we use a threshold $\tau$ to generate a confidence map $\mathcal{C}_{LC} = \mathbb{I}(\tilde{Y}^t_{LC} > \tau)$, where $\mathbb{I}$ is the indicator function.

**Height Pseudo-Label Generation.** Height pseudo-labels are generated by leveraging prior knowledge that only trees and buildings have height values. This is the first attempt to use ground information to correct height pseudo-labels, inspired by the empirical observations that the network achieves superior accuracy on the ground class early in the training stage. We refine height pseudo-labels using a ground mask $\mathcal{G}$ generated from the land cover mapping branch as $\mathcal{G} = \mathbb{I}(\tilde{Y}^t_{LC} = \text{Ground})$, and refine the height pseudo-labels by $\tilde{Y}^{t,refined}_H = \tilde{Y}^{t,ori}_H \cdot (1 - \mathcal{G})$. We also create a height consistency map $\mathcal{C}_H$:

$$\mathcal{C}_H = \mathbb{I}\left( \max\left( \frac{\tilde{Y}^{t,ori}_H}{\tilde{Y}^{t,aug}_H}, \frac{\tilde{Y}^{t,aug}_H}{\tilde{Y}^{t,ori}_H} \right) \leq \eta \right), \tag{2}$$

indicating that we consider predictions reliable if they remain stable under perturbations, where $\eta$ is the threshold.

**Target Domain Training.** The target domain training loss is:

$$\mathcal{L}_{target} = \frac{1}{N} \sum_{i=1}^{N} \left( \mathcal{L}_{CE}(\hat{Y}_{LC}^{t(i)}, \tilde{Y}_{LC}^{t(i)}) \cdot \mathcal{C}_{LC}^{(i)} + \mathcal{L}_{SmoothL1}(\hat{Y}_{H}^{t(i)}, \tilde{Y}_{H}^{t,refined(i)}) \cdot \mathcal{C}_{H}^{(i)} \right). \quad (3)$$

**Feature Constraint.** To alleviate overfitting, we adopt a frozen DINOv2 [71] encoder to supervise the student encoder's updates, inspired by Yang et al. [109]. Let $\mathbf{f}$ denote the features of the student encoder and $\mathbf{f}'$ denote the features of the frozen DINOv2 encoder. We utilize cosine similarity to constrain the feature updates with the loss defined as:

$$\mathcal{L}_{feat} = \begin{cases} 1 - \frac{\mathbf{f} \cdot \mathbf{f}'}{\|\mathbf{f}\|\|\mathbf{f}'\|} & \text{if } \frac{\mathbf{f} \cdot \mathbf{f}'}{\|\mathbf{f}\|\|\mathbf{f}'\|} < \epsilon \\ 0 & \text{otherwise} \end{cases}, \quad (4)$$

where $\epsilon$ is the threshold.

**Overall Loss.** The overall loss of the model is defined as the sum of the source domian training loss, target domain training loss, and feature alignment loss, each weighted by their respective coefficients. Formally, the overall loss is given by:

$$\mathcal{L}_{overall} = \mathcal{L}_{source} + \lambda_{target}\mathcal{L}_{target} + \lambda_{feat}\mathcal{L}_{feat}, \quad (5)$$

where $\lambda_{target}$ and $\lambda_{feat}$ are weighting coefficients that control the contribution of the target and feature alignment loss terms, respectively.

## 5 Experiments

**Evaluation Datasets & Experimental Setting.** We evaluate our synthetic dataset, SynRS3D, from various aspects. Table 2 (a) shows the real-world height estimation datasets, while Table 2 (b) lists the real-world land cover mapping datasets. In Section 5.1, we compare SynRS3D with other synthetic datasets under the source-only setting, a term commonly used in UDA to describe models trained solely on the source domain and directly applied to the target domain without adaptation. This setup highlights SynRS3D's smaller domain gap and its potential for direct usage in real-world scenarios. In Section 5.2, we investigate the advantages of SynRS3D for augmenting real-world data through fine-tuning and joint training. Specifically, in Section 5.3, to evaluate the effectiveness of RS3DAda, we divide the 11 height estimation datasets into two target domains: *Target Domain 1* includes 6 widely-used public datasets, while *Target Domain 2* contains 5 more challenging datasets to assess and contrast the generalization ability of SynRS3D with real datasets.

**Evaluation Metrics.** We employ Intersection over Union (IoU) and Mean Intersection over Union (mIoU) as the metrics to evaluate the model's performance for land cover mapping tasks. For height estimation, we use Mean Absolute Error (MAE), Root Mean Squared Error (RMSE), and accuracy metrics [18] denoted as $\delta$, along with our custom metric $F_1^{HE}$. Detailed metric definitions are in Appendix A.3.

**Implementation Details.** Unless specifically detailed, all experiments utilize the pre-trained DINOv2 [71] implemented with ViT-L [17] as the encoder, with DPT [74] serving as both the land cover mapping and height estimation decoders. The hyperparameters for the various experiments are provided in Appendix A.4.

Table 2: Datasets setup for the experiments on height estimation and land cover mapping. (a) **Height Estimation Datasets**: We detail 11 height estimation datasets categorized into two target domains. The first six datasets for training the model with real-world data are sourced from Europe and the United States as shown in Section 5.3. The remaining five datasets covering more challenging areas are characterized by: notable height mean&standard deviation, non-RGB channels, and varied regions outside of the US and EU, used for evaluation in Section 5.3. (b) **Land Cover Mapping Datasets**: We evaluate our method on five commonly used datasets covering diverse environments.

| Real-World Height Estimation Datasets | | | | |
|---|---|---|---|---|
| **Types** | **Datasets** | **Region** | **Height mean&std** | **Channel** |
| Target Domain 1 | Houston [106] | US | [3.07, 5.02] | RGB |
| | JAX [45] | US | [4.73, 9.02] | RGB |
| | OMA [45] | US | [2.37, 5.27] | RGB |
| | GeoNRW_Urban [5] | Germany | [2.46, 4.31] | RGB |
| | GeoNRW_Rural [5] | Germany | [2.03, 4.21] | RGB |
| | Potsdam [80] | Germany | [3.02, 5.68] | RGB |
| Target Domain 2 | ATL [13] | US | [8.40, 13.41] | RGB |
| | ARG [13] | Argentina | [3.90, 4.29] | RGB |
| | Nagoya [15] | Japan | [7.36, 11.84] | RGB |
| | Tokyo [15] | Japan | [15.73, 22.77] | RGB |
| | Vaihingen [80] | Germany | [2.36, 3.57] | NIR, G, B |

(a)

| Real-World Land Cover Mapping Datasets | | | |
|---|---|---|---|
| **Types** | **Datasets** | **Region** | **Categories** |
| Target Domain | OEM [101] | Global | 8 |
| | Vaihingen [80] | Germany | 6 |
| | Potsdam [80] | Germany | 6 |
| | JAX [45] | US | 6 |
| | OMA [45] | US | 6 |

(b)

## 5.1 Source-only Scenarios

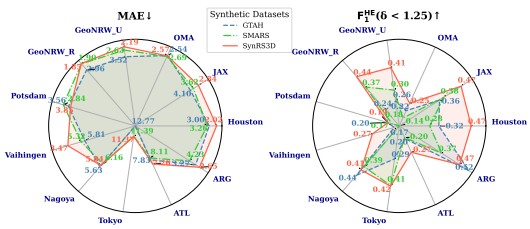

Figure 5: Source-only height estimation comparison of SynRS3D and other synthetic datasets showing in different metrics.

**Height Estimation.** We compared SynRS3D with other synthetic datasets in a source-only height estimation scenario. As shown in Figure 5, SynRS3D outperformed competitors SMARS [76] and GTAH [104] in 9 out of 11 real datasets. This superiority is attributed to the smaller domain gap and diversity of SynRS3D, as well as the precise calculation of heights within 3D software. In contrast, heights in SMARS and GTAH are not normalized, requiring additional algorithms for normalization, which introduces inherent noise in their height values.

**Land Cover Mapping.** We demonstrate the source-only capability of models trained on SynRS3D in the land cover mapping task. Table 3 compares SynRS3D with existing synthetic datasets, SMARS [76] and SyntheWorld [86]. Due to category inconsistencies, we present IoU and mIoU for shared categories including trees, buildings, and ground on the JAX [45], OMA [45], Vaihingen [80], and Potsdam [80] datasets. The model trained on SynRS3D achieves the best results across these four real datasets using only land cover labels, demonstrating the extraordinary compatibility of SynRS3D.

Table 3: Source-only land cover mapping performance on various real-world datasets.

| Datasets | JAX [45] | | | | OMA [45] | | | | Vaihingen [80] | | | | Potsdam [80] | | | |
|---|---|---|---|---|---|---|---|---|---|---|---|---|---|---|---|---|
| | Ground | Tree | Building | mIoU | Ground | Tree | Building | mIoU | Ground | Tree | Building | mIoU | Ground | Tree | Building | mIoU |
| SMARS [76] | 76.02 | 43.13 | 61.28 | 60.14 | 82.17 | 17.25 | 59.94 | 53.12 | 74.10 | 58.40 | 74.35 | 68.95 | 68.56 | 5.35 | 57.51 | 43.81 |
| SyntheWorld [86] | 74.63 | 54.74 | 64.18 | 64.52 | 81.29 | **45.83** | 56.56 | 61.23 | 72.69 | 68.09 | 75.67 | 72.15 | 69.09 | 32.49 | 55.88 | 52.49 |
| **SynRS3D** | **77.69** | **57.03** | **68.96** | **67.89** | **83.96** | 41.08 | **62.28** | **62.44** | **75.66** | **68.58** | **79.61** | **74.61** | **74.26** | **35.34** | **69.46** | **59.69** |

## 5.2 Combining SynRS3D with Real Data Scenarios

An important use of synthetic data is to augment real-world data. To demonstrate this capability of SynRS3D, we conducted two experiments. First, we trained models on SynRS3D and fine-tuned them on real data. Second, we combined SynRS3D with real data for joint training. We experimented with two different backbones: DINOv2 [71]+DPT [74] and DeepLabV2 [11]+ResNet101 [29]. Figure 6 (a) showcases SynRS3D's performance in the height estimation task on three city datasets: JAX [45],

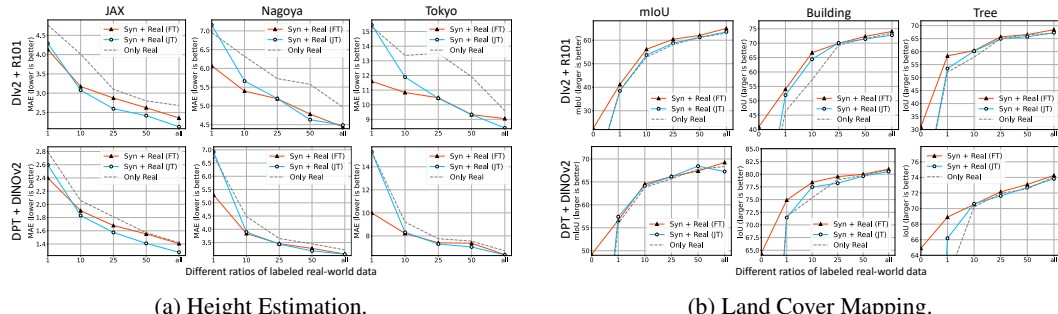

(a) Height Estimation.

(b) Land Cover Mapping.

Figure 6: Performance evaluation of combining SynRS3D with real data in (a) height estimation and (b) land cover mapping. Height estimation is evaluated on various real datasets, and land cover mapping is evaluated on OEM dataset, showing IoU for building, tree, and mIoU. FT: fine-tuning on real data after pre-training on SynRS3D, JT: joint training with SynRS3D and real data.

Nagoya [15], and Tokyo [15]. The results indicate that both approaches yield significant improvements when real data is scarce, with benefits diminishing as more real data is added. Additionally, the stronger the backbone, the smaller the improvement provided by SynRS3D, and vice versa. Figure 6 (b) illustrates SynRS3D's augmentation capability in the land cover mapping task on OEM [101] dataset, showing similar conclusions to the height estimation task.

## 5.3 Transfer SynRS3D to Real-World Scenarios

Table 4: Results of RS3DAda height estimation branch using DINOv2 [71] and DPT [74]. The experimental results are divided as follows: "Whole" denotes the evaluation results for the entire image. "High" signifies the experimental results for image regions above 3 meters. T.D.1 and T.D.2 correspond to *Target Domain 1* and *Target Domain 2*, respectively, as specified in Table 2. Avg. stands for the average value.

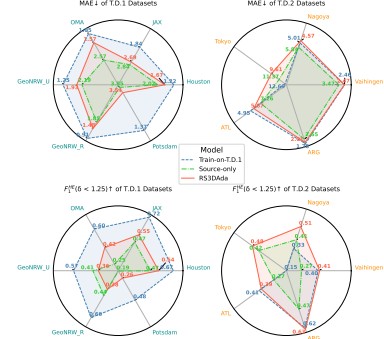

Figure 7: Results of RS3DAda height estimation branch for each dataset.

| Model | MAE ↓ | | RMSE ↓ | | Accuracy Metrics [18] ↑ | | | $F_1^{HE}$ ↑ | | |
|---|---|---|---|---|---|---|---|---|---|---|
| | Whole | High | Whole | High | $\delta < 1.25$ | $\delta < 1.25^2$ | $\delta < 1.25^3$ | $\delta < 1.25$ | $\delta < 1.25^2$ | $\delta < 1.25^3$ |
| Avg. T.D.1 | Whole | High | Whole | High | $\delta < 1.25$ | $\delta < 1.25^2$ | $\delta < 1.25^3$ | $\delta < 1.25$ | $\delta < 1.25^2$ | $\delta < 1.25^3$ |
| Train-on-T.D.1 | **1.272** | **3.363** | **2.381** | **4.329** | **0.379** | **0.463** | **0.510** | **0.617** | **0.710** | **0.742** |
| Source Only | 2.557 | 5.617 | 4.128 | 6.705 | 0.123 | 0.192 | 0.246 | 0.372 | 0.491 | 0.552 |
| **RS3DAda** | 2.148 | 4.921 | 3.593 | 6.024 | 0.185 | 0.258 | 0.318 | 0.418 | 0.554 | 0.623 |
| Avg. T.D.2 | Whole | High | Whole | High | $\delta < 1.25$ | $\delta < 1.25^2$ | $\delta < 1.25^3$ | $\delta < 1.25$ | $\delta < 1.25^2$ | $\delta < 1.25^3$ |
| Train-on-T.D.1 | 5.378 | 8.302 | 8.301 | 10.714 | 0.146 | 0.244 | 0.336 | 0.384 | 0.535 | 0.627 |
| Source Only | 6.117 | 8.923 | 9.221 | 11.443 | 0.125 | 0.223 | 0.312 | 0.365 | 0.514 | 0.601 |
| **RS3DAda** | **4.866** | **7.227** | **7.584** | **9.594** | **0.182** | **0.299** | **0.389** | **0.485** | **0.621** | **0.689** |

**Heigh Estimation Branch.** Table 4 shows the height estimation results for our RS3DAda model. "Whole" refers to the entire image, and "High" focuses on targets above 3 meters, such as trees and buildings. Using DINOv2 [71] and DPT [74] models, RS3DAda reduces the average MAE by 0.409 meters over the source-only approach across six datasets in *Target Domain 1*, though it is still exceeded by the models trained directly on these datasets. In *Target Domain 2*, RS3DAda outperforms models trained on real-world data, indicating its strong generalization under challenging scenarios, featuring diverse geographic regions and complex terrain characteristics. Figure 7 aligns with these results, showing RS3DAda's improvements on each dataset in *Target Domain 1* and *Target Domain 2*. This demonstrates SynRS3D's potential and the effectiveness of RS3DAda. However, the gap in *Target Domain 1* highlights the ongoing need to bridge the synthetic-to-real data gap, providing a benchmark for future UDA algorithm development in height estimation tasks.

**Land Cover Mapping Branch.** Table 5 presents the results of the RS3DAda model for the land cover mapping branch, evaluated using the OEM dataset. As shown, the RS3DAda method surpasses DAFormer by 1.94 in mIoU, indicating that the height branch positively impacts the land cover mapping performance. However, there remains a gap of 20.11 in mIoU compared to the Oracle

Table 5: Results of the RS3DAda land cover mapping branch on the OEM [101] dataset. All models are implemented with DINOv2 [71] and DPT [74].

| Model | Bareland | Rangeland | Developed | Road | Tree | Water | Agriculture | Buildings | mIoU |
|---|---|---|---|---|---|---|---|---|---|
| Source-only | 8.69 | 37.95 | **22.54** | **49.05** | 60.16 | 46.64 | 35.40 | **65.19** | 40.70 |
| DAFormer [35] | 12.54 | 41.16 | 10.88 | 43.88 | **62.56** | **77.55** | 62.62 | 59.10 | 46.29 |
| **RS3DAda** | **19.92** | **47.61** | 18.41 | 44.06 | 61.04 | 71.66 | **63.73** | 59.42 | **48.23** |
| Train-on-OEM | 50.04 | 59.10 | 58.18 | 65.39 | 73.07 | 83.65 | 76.36 | 80.88 | 68.34 |

model, suggesting significant room for improvement. Future research can build upon our method to make further advancements.

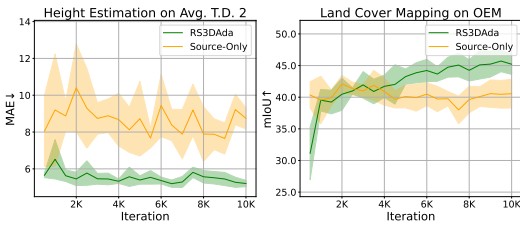

Figure 8: Performance of SynRS3D at the beginning of training for height estimation and land cover mapping branches, with and without the use of RS3DAda.

**Stabilizing Training on SynRS3D.** RS3DAda can regularize the training of synthetic data to prevent rapid overfitting at the beginning of the training, corresponding to the green section in Figure 8. Without RS3DAda, the model's evaluation results on the target domain fluctuate wildly during training in both height estimation and land cover mapping branches. This instability can lead to unreliable performance and poor generalization. RS3DAda ensures more consistent training, resulting in better model accuracy and stability.

Table 6: Comparison of RS3DAda with existing UDA methods AdaSeg and DADA. Supervision types: H for height maps, L for land cover labels. T.D.1 represents Target Domain 1, and T.D.2 represents Target Domain 2. V: Vaihingen [80]; P: Potsdam [80]; J: JAX [45]; O: OMA [45]. All models are implemented with DeepLabv2 and ResNet101. The datasets used for Tranin-on-Real are T.D.1 and OEM respectively.

**Comparison with Existing UDA.** We re-implemented AdaptSeg [91], DADA [94] and RS3DAda using DeepLabv2 [11] and ResNet101 [29] for fair comparison. Table 6 shows that RS3DAda outperformed both AdaptSeg and DADA in height estimation and land cover mapping tasks. Notably, when using weaker network architectures, the CNN encoder pre-trained on ImageNet fails to provide reliable RS image features and high accuracy for the ground category. As a result, the Feature Constraint and Ground-Guided Pseudo-Label Refinement in our RS3DAda cannot achieve their maximum effectiveness.

| Model | Supervision | Height Estimation (MAE)↓ | | Land Cover Mapping (mIoU)↑ | |
|---|---|---|---|---|---|
| | | Avg. T.D.1 | Avg. T.D.2 | OEM [101] | Avg. (V+P+J+O) |
| Source-Only | H, L | 3.911 | 7.419 | 17.42 | 39.61 |
| AdaptSeg [91] | L | - | - | 20.06 | 40.00 |
| DADA [94] | H+L | 3.615 | 6.997 | 21.24 | 46.44 |
| **RS3DAda** | H+L | **3.275** | **6.708** | **22.55** | **47.28** |
| Train-on-Real | H, L | 1.859 | 6.639 | 64.54 | 53.12 |

## 6   Conclusion and Discussion

In this work, we introduced SynRS3D, the largest synthetic remote sensing (RS) dataset, and RS3DAda, a multi-task unsupervised domain adaptation (UDA) method, designed to address the challenge of global 3D semantic reconstruction from single-view RS images. Our experiments on public datasets demonstrate the effectiveness of these tools in enhancing the use of synthetic data for RS research, setting a benchmark for 3D reconstruction from monocular RS images.

While SynRS3D offers a substantial contribution, there remains an appearance gap between synthetic and real-world data, potentially affecting real-world performance. Additionally, the dataset, though extensive, does not capture the full diversity of global cities, which could limit its generalizability. Future work will focus on reducing this gap and expanding the dataset's coverage to improve its robustness across different urban environments. By making these resources publicly available, we hope to stimulate further research and development in the field.

## Acknowledgments

This work was supported in part by JST FOREST Program Grant Number JPMJFR206S; Microsoft Research Asia; JSPS KAKENHI Grant Number 24KJ0652; the Next Generation AI Research Center of The University of Tokyo; the Japan Science and Technology Agency SPRING Program (JST SPRING) Grant Number JPMJSP2108; and RIKEN Junior Research Associate (JRA) Program.

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

# Appendix



## A   Technical Supplements

In this technical supplement, we provide detailed insights and additional results to support our main paper. Section A.1 outlines the generation process of the SynRS3D dataset, including the tools and plugins used. It also covers the licenses for these plugins. Section A.2 discusses the data sources and licenses of the existing real-world datasets utilized in our experiments. Section A.3 elaborates on the evaluation metrics for different tasks, including the proposed $F_1^{HE}$ metric specifically designed for remote sensing height estimation tasks. Section A.4 describes the experimental setup and the selection of hyperparameters for the RS3DAda method. Section A.5 presents the ablation study results and analysis for the RS3DAda method. Section A.6 provides supplementary experimental results combining SynRS3D and real data scenarios, complementing Section 5.2 of the main paper. Section A.7 showcases the qualitative visual results of RS3DAda on various tasks. Section A.8 details the generation process and samples of building change detection annotations in SynRS3D, as well as the evaluation results of the source-only scenario on different real datasets. Section A.9 highlights the performance of models trained on the SynRS3D dataset using RS3DAda in the critical application of disaster mapping in remote sensing.

### A.1   Detailed Generation Workflow of SynRS3D

The generation workflow of SynRS3D involves several key steps, from initializing sensor and sunlight parameters to generating the layout, geometry, and textures of the scene. This comprehensive process ensures that the generated SynRS3D mimics real-world remote sensing scenarios with high fidelity.

The main steps of the workflow are as follows:

- **Initialization:** Set up the sensor and sunlight parameters using uniform and normal distributions to simulate various conditions.
- **Layout Generation:** Define the grid and terrain parameters to create diverse urban and natural environments.
- **Geometry Generation:** Specify the characteristics of roads, rivers, buildings, and vegetation, ensuring realistic representations.
- **Texture Generation:** Use advanced models like GPT-4 [2] and Stable Diffusion [78] to generate realistic textures for different categories of land cover.
- **Scene Construction and Processing:** Assemble the scene with all generated components and apply textures to create visually accurate post-event and pre-event images.
- **Outlier Filtering:** Filter outliers based on height maps to ensure the quality and reliability of the dataset.

The detailed algorithm for this workflow is provided in Algorithm 1. The development process of SynRS3D is based on Blender 3.4, where we utilized and modified various community add-ons to facilitate the generation of SynRS3D. A comprehensive list of all the add-ons used during our development process is presented in Table 7.

Table 7: List of Blender add-ons used in the SynRS3D.

| Name | Author | Version | License | URL |
|---|---|---|---|---|
| Realtime River Generator | specoolar | 1.1 | RF | https://blendermarket.com/products/river-generator |
| Next Street | Next Realm | 2.0 | RF | https://blendermarket.com/products/next-street |
| Objects Replacer | Georeality Design | 1.06 | GPL | https://blendermarket.com/products/objects-replacer/docs |
| Albero | Greenbaburu | 0.3 | RF | https://blendermarket.com/products/albero---geometry-nodes-powered-tree-generator |
| Hira Building Generator | HiranojiStore | 0.9 | RF | https://blendermarket.com/products/hira-building-generator |
| Procedural Building Generator | Isak Waltin | 1.2.1 | CC-BY 4.0 | https://blendermarket.com/products/building-gen |
| Pro Atmo | Contrastrender | 1.0 | GPL | https://blendermarket.com/products/pro-atmo |
| Modular Buildings Creator | PH Felix | 1.0 | RF | https://blendermarket.com/products/modular-buildings-creator |
| Next Trees | Next Realm | 2.0 | RF | https://blendermarket.com/products/next-trees |
| SceneCity | Arnaud | 1.9.3 | RF | http://www.cgchan.com/store/scenecity |
| Flex Road Generator | EasyNodes | 1.1.0 | RF | https://www.cgtrader.com/3d-models/scripts-plugins/modelling/blender-mesh-curve-to-road |
| Buildify | Pavel Oliva | 1.0 | RF | https://paveloliva.gumroad.com/l/buildify |

### A.2   License and Data Source of Real-World Datasets

The licenses and data sources for the real-world datasets used for evaluation and training in this work are shown in Table 8. For the Potsdam, Vaihingen, GeoNRW, Nagoya, and Tokyo datasets, we used the dsm2dtm [2] algorithm to convert them to normalized Digital Surface Model (nDSM), since they only provide Digital Surface

---

[2]https://github.com/seedlit/dsm2dtm

---

**Algorithm 1** Generation Workflow of SynRS3D

---

1: **Initialize Parameters**
2: $\mathcal{S} \leftarrow \{\text{azimuth} \sim U(a_1, a_2), \text{look\_angle} \sim \mathcal{N}(\mu_1, \sigma_1), \text{GSD} \sim \mathcal{N}(\mu_2, \sigma_2)\}$ # $\mathcal{S}$: Sensor parameters
3: $\mathcal{L} \leftarrow \{\text{elevation} \sim U(e_1, e_2), \text{intensity} \sim U(i_1, i_2), \text{color} \sim [U(c_1, c_2), U(c_1, c_2), U(c_1, c_2)]\}$ # $\mathcal{L}$: Sunlight parameters
4: **Generate Layout**
5: $\mathcal{G} \leftarrow \{\text{district\_num} \sim \text{randint}(d_1, d_2), \text{district\_size} \sim \text{randint}(s_1, s_2), \text{obj\_density} \sim U(o_1, o_2)\}$ # $\mathcal{G}$: Grid parameters
6: $\mathcal{T} \leftarrow \{\text{flat\_area} \sim U(f_1, f_2), \text{mountain\_area} \sim U(m_1, m_2), \text{sea\_area} \sim U(s_1, s_2), \text{tree\_density} \sim U(t_1, t_2)\}$ # $\mathcal{T}$: Terrain parameters
7: **Generate Geometry**
8: $\mathcal{R} \leftarrow \{\text{river\_num} \sim \text{randint}(r_1, r_2), \text{road\_num} \sim \text{randint}(r_3, r_4), \text{width} \sim U(w_1, w_2)\}$ # $\mathcal{R}$: Road and River parameters
9: $\mathcal{B} \leftarrow \{\text{height} \sim U(h_1, h_2), \text{type} \in \text{select(types)}, \text{roof\_angle} \sim U(ra_1, ra_2)\}$ # $\mathcal{B}$: Building parameters
10: $\mathcal{V} \leftarrow \{\text{trunk} \sim \text{Sample\_Curve}(), \text{branch\_num} \sim \text{randint}(b_1, b_2), \text{leaf\_num} \sim \text{randint}(l_1, l_2)\}$ # $\mathcal{V}$: Tree parameters
11: **Generate Textures**
12: $\mathcal{C} \leftarrow \{\text{Rangeland, Agricultural Land, Bareland, Developed Space, Road, Roof}\}$ # $\mathcal{C}$: Texture categories
13: **for** category $\in \mathcal{C}$ **do**
14:     texture\_prompts $\leftarrow$ GPT-4(category)
15:     textures[category] $\leftarrow$ Stable\_Diffusion(texture\_prompts)
16: **end for**
17: **Construct Scene**
18: $\mathcal{P}_s \leftarrow \text{create\_scene}(\mathcal{S} \cup \mathcal{L} \cup \mathcal{G} \cup \mathcal{T} \cup \mathcal{R} \cup \mathcal{B} \cup \mathcal{V})$ # $\mathcal{P}_s$: Post-event scene
19: $\mathcal{Q}_s \leftarrow \text{remove\_buildings}(\text{copy}(\mathcal{P}_s), U(rb_1, rb_2))$ # $\mathcal{Q}_s$: Pre-event scene
20: **Process Scene**
21: $\mathcal{P}_t \leftarrow \text{apply\_textures}(\mathcal{P}_s, \text{textures})$ # $\mathcal{P}_t$: Post-event scene with textures
22: $\mathcal{Q}_t \leftarrow \text{apply\_textures}(\mathcal{Q}_s, \text{textures})$ # $\mathcal{Q}_t$: Pre-event scene with textures
23: $\mathcal{P}_r \leftarrow \text{render\_rgb}(\mathcal{P}_t)$ # $\mathcal{P}_r$: Post-event RGB image
24: $\mathcal{Q}_r \leftarrow \text{render\_rgb}(\mathcal{Q}_t)$ # $\mathcal{Q}_r$: Pre-event RGB image
25: $\mathcal{P}_l \leftarrow \text{generate\_land\_cover}(\mathcal{P}_t)$ # $\mathcal{P}_l$: Post-event land cover mapping
26: $\mathcal{P}_h \leftarrow \text{generate\_height\_map}(\mathcal{P}_t)$ # $\mathcal{P}_h$: Post-event height map
27: $\mathcal{P}_b \leftarrow \text{generate\_building\_mask}(\mathcal{P}_t)$ # $\mathcal{P}_b$: Post-event building mask
28: $\mathcal{Q}_b \leftarrow \text{generate\_building\_mask}(\mathcal{Q}_t)$ # $\mathcal{Q}_b$: Pre-event building mask
29: $\mathfrak{C} \leftarrow \text{subtract\_masks}(\mathcal{P}_b, \mathcal{Q}_b)$ # $\mathfrak{C}$: Building change detection mask
30: **Filter Outliers** # Input: $\mathcal{P}_h, H_T, H_m, H_s$; Output: $\mathcal{F}_{\mathcal{P}_h}$ (Filtered height map list)
31: $H_T \leftarrow$ threshold value # Set the height threshold value
32: $H_m \leftarrow$ minimum threshold # Set the minimum proportion threshold
33: $H_s \leftarrow$ steepness value # Set the steepness value for the sigmoid function
34: $\mathcal{F}_{\mathcal{P}_h} \leftarrow \emptyset$ # Initialize the filtered height map set
35: **for** each $n \in \mathcal{P}_h$ **do**
36:     $a \leftarrow \text{read\_image}(n)$ # Read the height map as a numpy array
37:     $T_p \leftarrow \text{total\_pixels}(a)$ # Calculate the total number of pixels
38:     $A_t \leftarrow \text{count\_above\_threshold}(a, H_T)$ # Count the number of pixels above the threshold
39:     $P_c \leftarrow \frac{A_t}{T_p}$ # Calculate the proportion of pixels above the threshold
40:     **if** $P_c \geq H_m$ **then**
41:         $\mathcal{F}_{\mathcal{P}_h} \leftarrow \mathcal{F}_{\mathcal{P}_h} \cup \{n\}$ # If proportion is above minimum threshold, add to filtered list
42:     **else**
43:         $Pr \leftarrow \frac{1}{1+e^{-H_s \cdot (P_c - H_m)}}$ # Calculate the probability using a sigmoid function
44:         **if** $\text{random}() < Pr$ **then**
45:             $\mathcal{F}_{\mathcal{P}_h} \leftarrow \mathcal{F}_{\mathcal{P}_h} \cup \{n\}$ # Add to filtered list based on probability
46:         **end if**
47:     **end if**
48: **end for**
49: **Output** # Output: SynRS3D dataset
50: $\{\mathcal{F}_{\mathcal{P}_r}, \mathcal{F}_{\mathcal{Q}_r}, \mathcal{F}_{\mathcal{P}_l}, \mathcal{F}_{\mathcal{P}_h}, \mathcal{F}_{\mathfrak{C}}\}$ # $\mathcal{F}_{\mathcal{P}_r}$: Filtered post-event RGB images, $\mathcal{F}_{\mathcal{Q}_r}$: Filtered pre-event RGB images, $\mathcal{F}_{\mathcal{P}_l}$: Filtered post-event land cover mappings, $\mathcal{F}_{\mathcal{P}_h}$: Filtered post-event height maps, $\mathcal{F}_{\mathfrak{C}}$: Filtered building change detection masks

---

Table 8: The data source and license of real-world height estimation datasets used in this work.

| | Datasets | Data Source | License/Conditions of Use |
|---|---|---|---|
| **Real-World Datasets** | | | |
| **Types** | **Datasets** | **Data Source** | **License/Conditions of Use** |
| Target Domain 1 | Houston [106] | Data Fusion Contest 2018 | Creative Commons Attribution |
| | JAX [45] | Data Fusion Contest 2019 | Creative Commons Attribution |
| | OMA [45] | Data Fusion Contest 2019 | Creative Commons Attribution |
| | GeoNRW_Urban [5] | GeoNRW | Creative Commons Attribution |
| | GeoNRW_Rural [5] | GeoNRW | Creative Commons Attribution |
| | Potsdam [80] | ISPRS | Research Purposes Only, No Redistribution |
| Target Domain 2 | ATL [13] | Overhead Geopose Challenge | Creative Commons Attribution |
| | ARG [13] | Overhead Geopose Challenge | Creative Commons Attribution |
| | Nagoya [15] | NTT DATA Corporation and Inc. DigitalGlobe | End User License Agreement |
| | Tokyo [15] | NTT DATA Corporation and Inc. DigitalGlobe | End User License Agreement |
| | Vaihingen [80] | ISPRS | Research Purposes Only, No Redistribution |

Model (DSM). We will release the processed real-world datasets upon acceptance, provided that the original datasets are allowed to be redistributed and are intended for non-commercial use.

## A.3 Evaluation Metrics

We utilized several metrics to ensure a comprehensive assessment of model performance when evaluating land cover mapping and height estimation tasks. In the following parts, we provide a detailed explanation and formulation of adopted metrics.

### A.3.1 Land Cover Mapping

**Intersection over Union (IoU)**  Intersection over Union (IoU) is a common evaluation metric used in image segmentation tasks. It measures the overlap between the predicted segmentation and the ground truth segmentation. The IoU for a single class is defined as:

$$\text{IoU} = \frac{|A \cap B|}{|A \cup B|},\tag{6}$$

where $A$ is the set of predicted pixels and $B$ is the set of ground truth pixels.

**Mean Intersection over Union (mIoU)**  mIoU extends IoU to multiple classes by averaging the IoU values of all classes. If there are $N$ classes, mIoU is calculated as:

$$\text{mIoU} = \frac{1}{N} \sum_{i=1}^{N} \text{IoU}_i,\tag{7}$$

where $\text{IoU}_i$ is the IoU for class $i$. This metric provides a single scalar value that summarizes the segmentation performance across all classes.

### A.3.2 Height Estimation

**Mean Absolute Error (MAE)**  Mean Absolute Error (MAE) measures the average magnitude of the errors between the predicted heights and the true heights. Suppose the ground truth heights are $Y$ and the predicted heights are $\hat{Y}$, and $n$ is the number of samples. It is defined as:

$$\text{MAE} = \frac{1}{n} \sum_{i=1}^{n} |Y_i - \hat{Y}_i|.\tag{8}$$

**Root Mean Squared Error (RMSE)**  Root Mean Squared Error (RMSE) measures the square root of the average squared differences between predicted heights and actual heights. It is defined as:

$$\text{RMSE} = \sqrt{\frac{1}{n} \sum_{i=1}^{n} (Y_i - \hat{Y}_i)^2}.\tag{9}$$

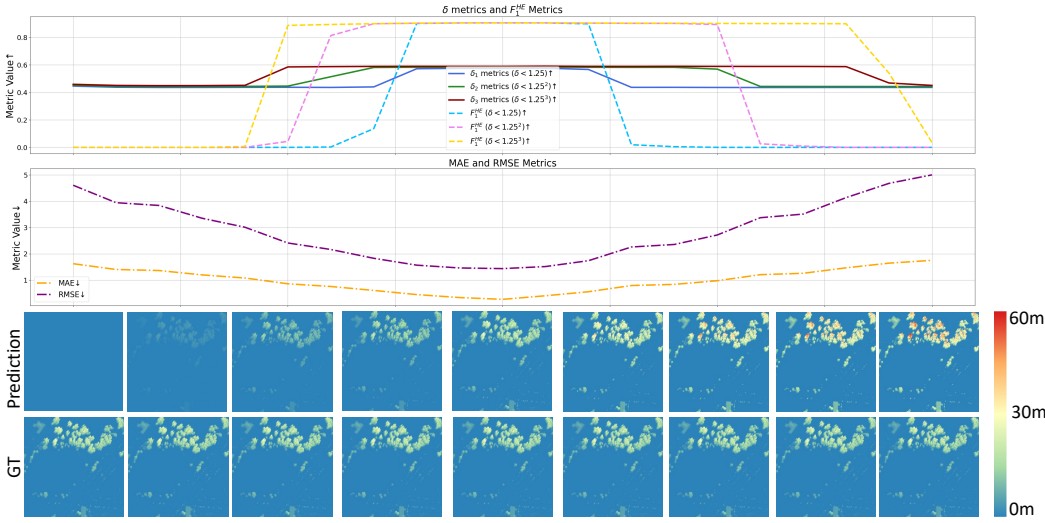

Figure 9: Comparison of proposed $F_1^{HE}$ metric and other metrics.

**Accuracy Metric** This metric, also called $\delta$ metric from early depth estimation work [18], evaluates the proportion of height predictions that fall within a certain ratio of the true heights. We use $\delta$ to represent a maxRatio map, which is calculated as follows:

$$\delta = \max\left(\frac{\hat{Y}}{Y}, \frac{Y}{\hat{Y}}\right). \tag{10}$$

Then, threshold values $\eta$ are used to measure the accuracy of the height predictions, the values of $\eta$ are usually $1.25, 1.25^2, 1.25^3$.

**F1 Score for Height Estimation** $(F_1^{HE})$ The $F_1^{HE}$ score innovatively applies the F1 score, typically used in classification, to the regression task of height estimation. This metric emphasizes both precision and recall in estimating significant heights. The $F_1^{HE}$ score balances precision and recall for height predictions above a significance threshold $T$ (e.g., 1 meter). The maxRatio is calculated as in equation 10. True Positives (TP), False Positives (FP), and False Negatives (FN) are identified as follows:

$$TP = \sum\left(\left(\hat{Y} > T \wedge Y > T\right) \wedge (\delta < \eta)\right), \tag{11}$$

$$FP = \sum\left(\hat{Y} > T \wedge Y \leq T\right), \tag{12}$$

$$FN = \sum\left(\hat{Y} \leq T \wedge Y > T\right), \tag{13}$$

where the values of $\eta$ are usually $1.25, 1.25^2, 1.25^3$. Precision, Recall, and $F_1^{HE}$ are then calculated as:

$$Precision = \frac{TP}{TP + FP}, \tag{14}$$

$$Recall = \frac{TP}{TP + FN}, \tag{15}$$

$$F_1^{HE} = 2 \times \frac{Precision \times Recall}{Precision + Recall}. \tag{16}$$

Our motivation for proposing a new metric for height estimation arises from observing that existing metrics such as MAE, RMSE, and $\delta$ metrics, which are derived from depth estimation tasks, do not consider the unique characteristics of height estimation in remote sensing images. Specifically, a significant portion of the remote sensing images can be occupied by ground classes, leading to an abundance of zero height values in the ground truth. This imbalance impedes the evaluation of model performance when using traditional depth estimation metrics.

Table 9: Settings for RS3DAda experimental hyperparameters.

| Category | Parameter | Value |
|---|---|---|
| Statistical Image Translation [1] | Fourier Domain Adaptation (FDA) | beta_limit= 0.01 |
| | Histogram Matching (HM) | blend_ratio= [0.8, 1.0] |
| | Pixel Distribution Adaptation (PDA) | blend_ratio= [0.8, 1.0], transform_type="standard" |
| Strong Augmentation | ClassMix | $|\mathcal{C}|/2$ |
| | ColorJitter [2] | $p = 0.8$ |
| | GaussianBlur [3] | $p = 0.5$ |
| Pseudo Label Generation | Land Cover Confidence Threshold ($\tau$) | 0.95 |
| | Height Map Consistency Threshold ($\eta$) | 1.55 |
| Optimization | Optimizer | AdamW |
| | Encoder Learning Rate ($lr$) | $1 \times 10^{-6}$ |
| | Decoder Learning Rate | $10 \times lr$ |
| | Weight Decay | $5 \times 10^{-4}$ |
| | Batch Size | 2 |
| | Iterations | $40,000$ |
| | Warmup Steps | $1,500$ |
| | Warmup Mode | Linear |
| | Decay Mode | Polynomial |
| | EMA ($\alpha$) | 0.99 |
| Loss Function | Feature Loss Threshold ($\epsilon$) | 0.8 |
| | Weighting Coefficient for Target Loss ($\lambda_{target}$) | 1 |
| | Weighting Coefficient for Feature Loss ($\lambda_{feat}$) | 1 |

[1] https://albumentations.ai/docs/api_reference/augmentations/domain_adaptation/

[2] https://albumentations.ai/docs/api_reference/augmentations/transforms/#albumentations.augmentations.transforms.ColorJitter

[3] https://albumentations.ai/docs/api_reference/augmentations/blur/transforms/#albumentations.augmentations.blur.transforms.GaussianBlur

As illustrated in Figure 9, when a network predicts all values as 0 meters or predicts the height of trees and buildings as twice their ground truth (30 meters to 60 meters), metrics like MAE, RMSE, and $\delta$ still indicate highly competitive accuracy. This is not reasonable because these metrics average the correct predictions of a large number of ground pixels. However, in height estimation tasks, the accuracy of predictions for objects with height is crucial. Our proposed $F_1^{HE}$ metric specifically addresses this issue by focusing on the accuracy of height predictions for objects higher than 1 meter. As shown, in both extreme cases, the F1 score is 0, reflecting the poor performance correctly. This metric better aligns with the objectives of the height estimation task. In practice, most images in height estimation datasets contain objects with heights exceeding 1 meter, so we skip the $F_1^{HE}$ calculation for images that only contain ground pixels.

This comprehensive evaluation framework ensures that height estimation models are assessed on both overall error rates and the ability to accurately predict significant height values in remote sensing images.

### A.4 Experimental Setting for RS3DAda

For the real-world datasets used in our experiments, we split each dataset into a 3:1 ratio for training and testing. In the RS3DAda experiments, we use random cropping of size 392 to ensure the dimensions are multiples of 14. The training batch size is set to 2, with each batch consisting of one labeled synthetic image from SynRS3D and one unlabeled image from the target domain training set.

Additionally, in RS3DAda, the teacher model is updated using Exponential Moving Average (EMA) of the student model parameters as follows:

$$\theta_t \leftarrow \alpha\theta_t + (1 - \alpha)\theta_s, \tag{17}$$

where $\theta_t$ represents the teacher model parameters, $\theta_s$ represents the student model parameters, and $\alpha$ is the EMA decay factor.

For detailed experimental parameters, please refer to Table 9.

### A.5 Ablation Studies of RS3DAda

In this section, we mainly conduct ablation experiments on the three key modules of RS3DAda: 1) the ground mask, 2) height map consistency, and 3) feature constraints. Additionally, we performed ablation studies on different mixing strategies in the strong augmentation of the target domain and the setting of the number of categories in the land cover branch. The evaluation dataset for height estimation experiments is *Target Domain 2*. For land cover mapping experiments, we employed the OEM [101] dataset for evaluation.

Table 10 presents the ablation study results for the RS3DAda method. Specifically, using DINOv2 [71] and DPT [74], we find that all three modules are important for height estimation, with the ground mask and height

Table 10: Ablation experiments of two types of network structures with our key modules, which were introduced in the RS3DADa section. MAE and $F_1^{HE}$ serve as evaluation metrics for the height estimation tasks, and IoU is used for the land cover mapping tasks.

| # | Model | Ground Mask | Height Consistency | Feature Constraint | Height Estimation | | | Land Cover Mapping |
|---|---|---|---|---|---|---|---|---|
| | | | | | MAE ↓ | $F_1^{HE}$ ($\delta < 1.25$) ↑ | mIoU ↑ | |
| 1 | DPT+DINOv2 | – | – | – | 6.117 | 0.365 | | 42.60 |
| 2 | DPT+DINOv2 | ✓ | – | – | 5.652 | 0.423 | | 44.05 |
| 3 | DPT+DINOv2 | ✓ | ✓ | – | 5.253 | 0.425 | | 44.75 |
| 4 | DPT+DINOv2 | ✓ | – | ✓ | 5.578 | 0.439 | | 42.93 |
| 5 | DPT+DINOv2 | – | ✓ | ✓ | 5.384 | 0.461 | | 46.67 |
| 6 | DPT+DINOv2 | ✓ | ✓ | ✓ | **4.886** | **0.485** | | **48.23** |
| 7 | DLv2+R101 | – | – | – | 7.419 | 0.318 | | 17.42 |
| 8 | DLv2+R101 | ✓ | ✓ | ✓ | 6.959 | 0.316 | | 18.89 |
| 9 | DLv2+R101 | ✓ | ✓ | – | **6.708** | **0.352** | | **22.55** |

consistency being particularly crucial. For instance, in Experiments 1 and 2, adding the ground mask reduces MAE from 6.117 to 5.652 and increases $F_1^{HE}$ from 0.365 to 0.423. Adding height consistency in Experiment 3 further improves performance, reducing MAE to 5.253 and increasing $F_1^{HE}$ to 0.425. The feature constraint, shown in Experiment 4, also contributes to improvements, though its impact is less significant. When all three modules are used together in Experiment 6, the best results are achieved with a MAE of 4.886, $F_1^{HE}$ of 0.485, and mIoU of 48.23. For land cover mapping, height consistency is essential. Without it, the model relies on land cover confidence for height regression, which is often insufficient. This lack of confidence in the pseudo labels for the height branch hinders the improvement of the height estimation branch, subsequently affecting the land cover branch. These results indicate that both branches support each other, and inadequate learning in one branch negatively impacts the other.

Interestingly, with the weaker network combination of DeepLabv2 [11] and ResNet101 [29] (Experiments 7-9), the feature constraint is ineffective. This is because the ImageNet-pretrained feature extractor, trained on natural images, does not generalize well to synthetic remote sensing data, unlike DINOv2's self-supervised pretraining on diverse datasets. Aligning features with the ImageNet-pretrained extractor hinders learning from synthetic data due to the significant domain gap. This demonstrates our method's effectiveness in leveraging DINOv2's features as a constraint.

Table 11: Comparison of mixing strategies and number of classes.

| Mix Strategy / #Class | Height Estimation | | Land Cover Mapping |
|---|---|---|---|
| | MAE ↓ | $F_1^{HE}$ ($\delta < 1.25$) ↑ | mIoU ↑ |
| **Mix Strategy** | | | |
| CutMix [114] | 4.966 | 0.475 | 47.34 |
| ClassMix [70] | **4.886** | **0.485** | **48.23** |
| **#Classes** | | | |
| 3 | 5.136 | 0.425 | - |
| 8 | **4.886** | **0.485** | - |

We also explored the impact of two different mix strategies and the number of land cover classes on the RS3DADa method. As shown in Tab. 11, ClassMix has a slight advantage over CutMix in both tasks. Regarding the number of land cover classes, we found that using all 8 land cover classes outperforms using only 3 classes (ground, tree, building). This improvement is likely because land cover mapping, being a segmentation task, benefits from a more detailed and discrete representation of features. In contrast, height estimation, which is a regression task, relies on continuous features. By having a finer label space in the classification branch, we can better align the segmentation and regression tasks, reducing the discrepancy between them.

## A.6 Additional Height Estimation Results in Combining SynRS3D and Real Data Scenarios

In the Section 5.2 of the main paper, we present height estimation results for three datasets. Here, we provide the remaining results for seven additional datasets. These results further demonstrate the efficacy of combining SynRS3D with real data across different environments for fine-tuning and joint training. Figures 10 and 11 showcase the performance across these additional datasets, following the same evaluation methodology as described in Section 5.2 of the main paper. These extended results support the main paper's conclusions, demonstrating that both fine-tuning on real data after pre-training on SynRS3D (FT) and joint training with

SynRS3D and real data (JT) significantly enhance model performance, especially when real data is limited. This underscores the importance of SynRS3D in complementing existing datasets and boosting model performance.

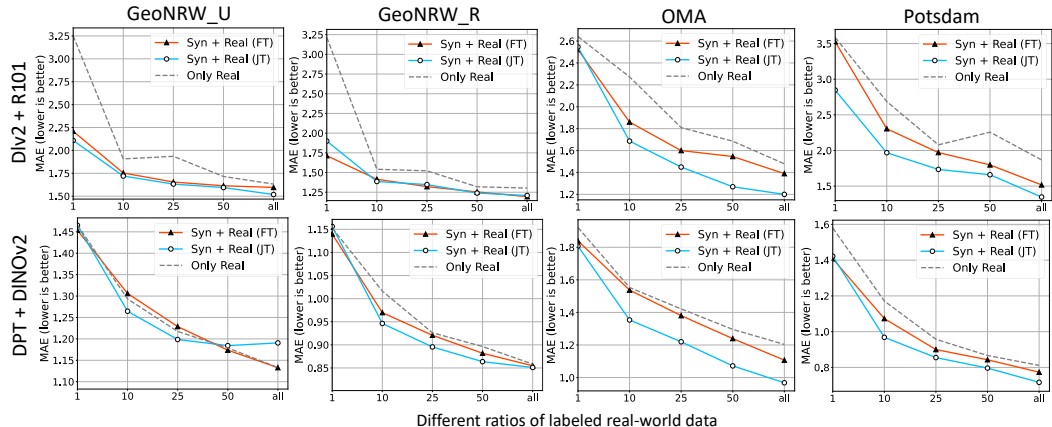

Figure 10: Additional performance evaluation on *Target Domain 1* datasets of combining SynRS3D with real data on height estimation task. FT: fine-tuning on real data after pre-training on SynRS3D, JT: joint training with SynRS3D and real data.

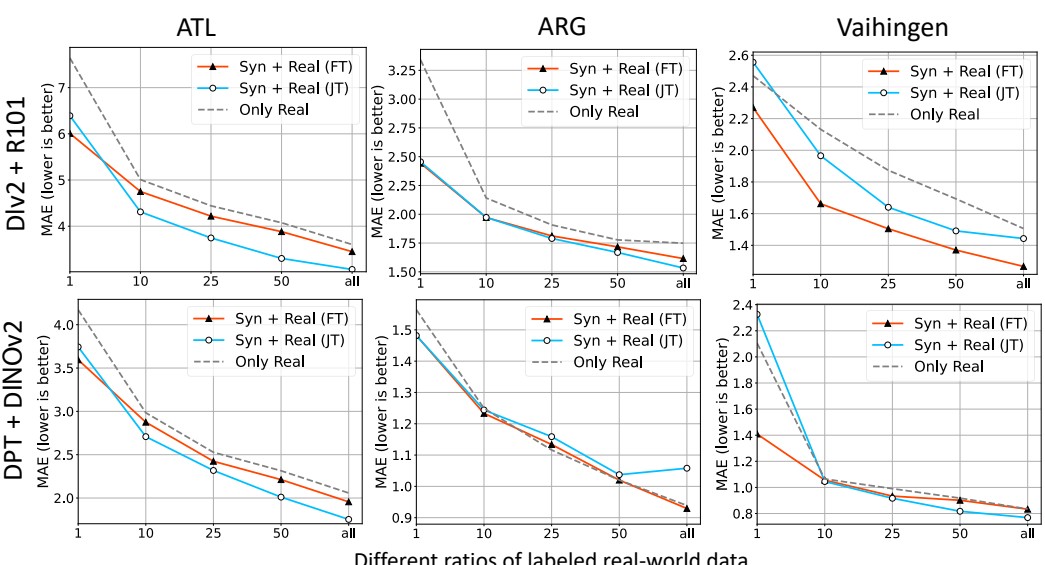

Figure 11: Additional performance evaluation on *Target Domain 2* datasets of combining SynRS3D with real data on height estimation task. FT: fine-tuning on real data after pre-training on SynRS3D, JT: joint training with SynRS3D and real data.

## A.7 Qualitative Results of RS3DAda

Figure 12 shows the qualitative results for the height estimation task. We can observe that the height predictions from the RS3DAda model are closer to the ground truth and have more complete edges. In contrast, the source-only model tends to overestimate height values and produces more incomplete edges. Although the model trained on *Target Domain 1* uses real data, it struggles to generalize to *Target Domain 2* due to its training data being limited to commonly available public datasets from European and American regions, which are unbalanced. As shown, its predicted heights are often underestimated. Figure 13 presents the qualitative results for the land cover mapping task. The RS3DAda model demonstrates exceptional performance in categories such as agricultural land, rangeland, and bare land, which aligns with our quantitative experimental results. However, it has some limitations in categories like roads and developed space, indicating that there is still significant room

for improvement in domain adaptation research for the SynRS3D dataset in the area of land cover mapping. This marks the first time in the field of remote sensing that synthetic data alone can achieve a high level of visual interpretation consistency with the ground truth. We hope that the RS3DAda method and the SynRS3D dataset can serve as benchmarks to further advance research in this direction. Figure 14 shows additional 3D reconstruction results in developing countries. These results are derived from using models trained on SynRS3D with RS3DAda to infer monocular satellite image tiles from Bing Satellite[3] and HereWeGo Satellite[4]. These 3D reconstruction areas cover between 3.2 square kilometers and 12.85 square kilometers, with a ground sample distance (GSD) of 0.35 meters.

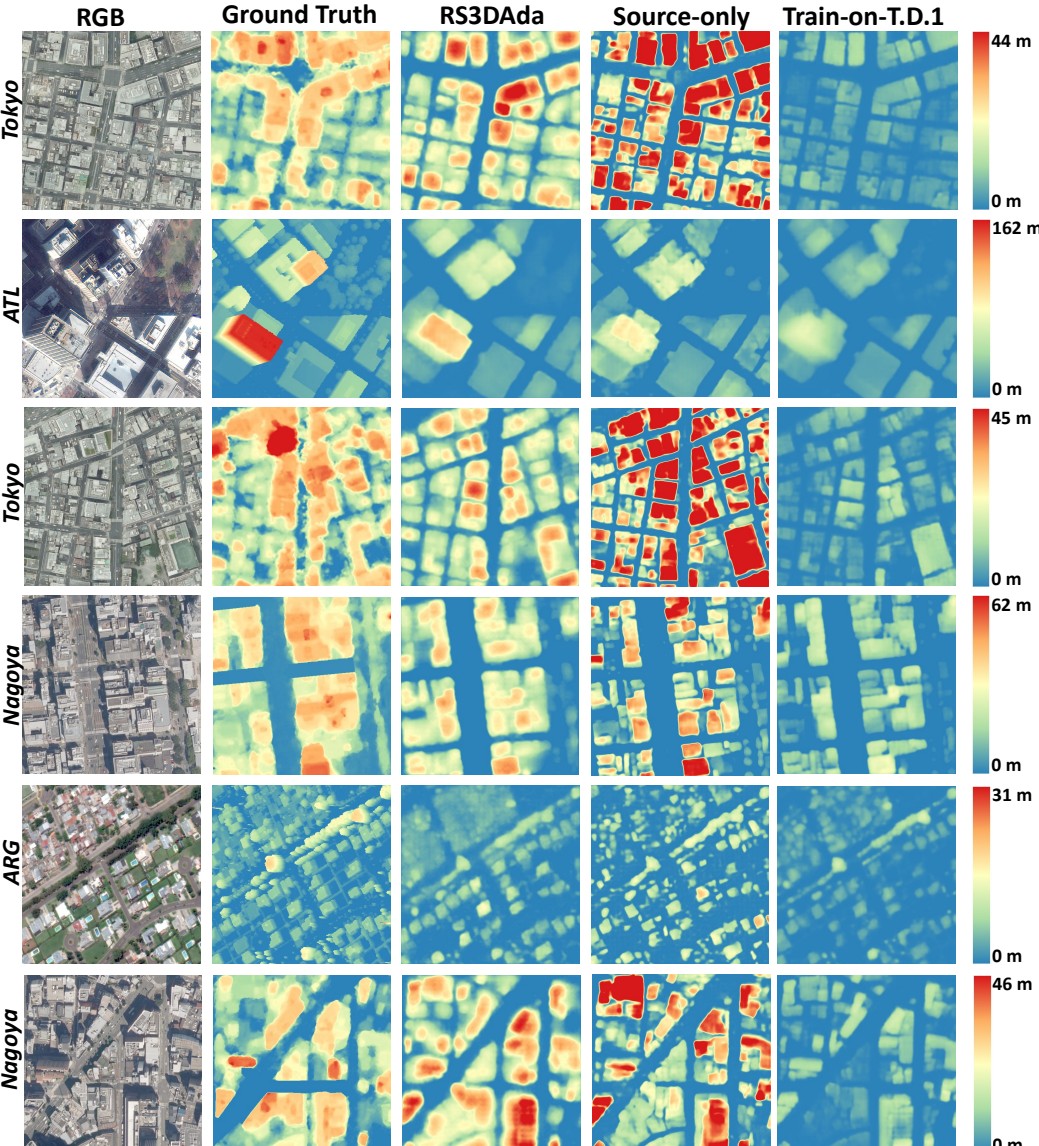

Figure 12: Qualitative results of height estimation task on *Target Domain 2* using the RS3DAda model, the source-only model, and the model trained on *Target Domain 1*. Satellite RGB images form Tokyo and Nagoya: © 2018 NTT DATA Corporation and Inc. DigitalGlobe.

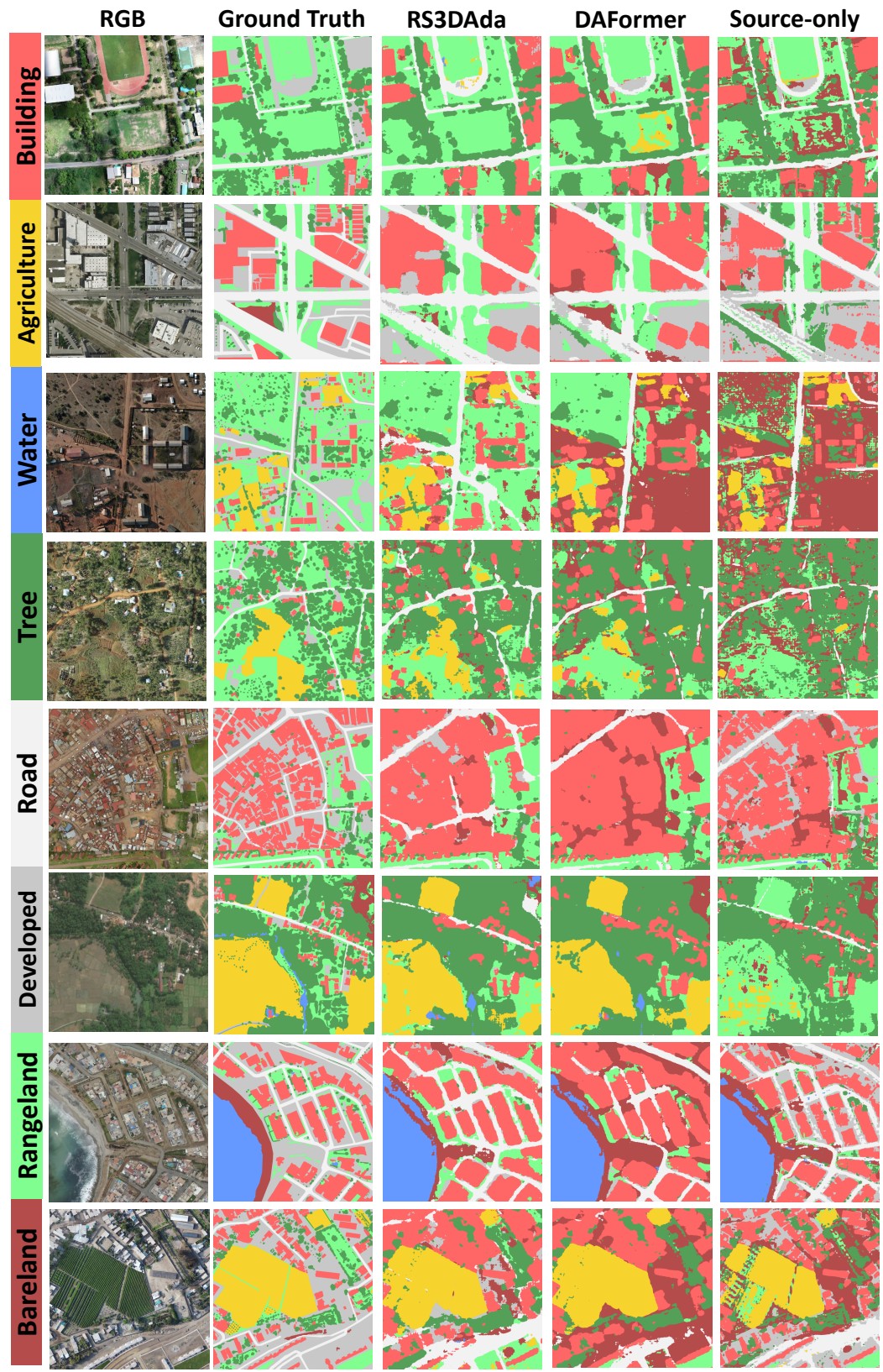

Figure 13: Qualitative results of land cover mapping task on OEM dataset using the RS3DAda model, the source-only model, and DAFormer.

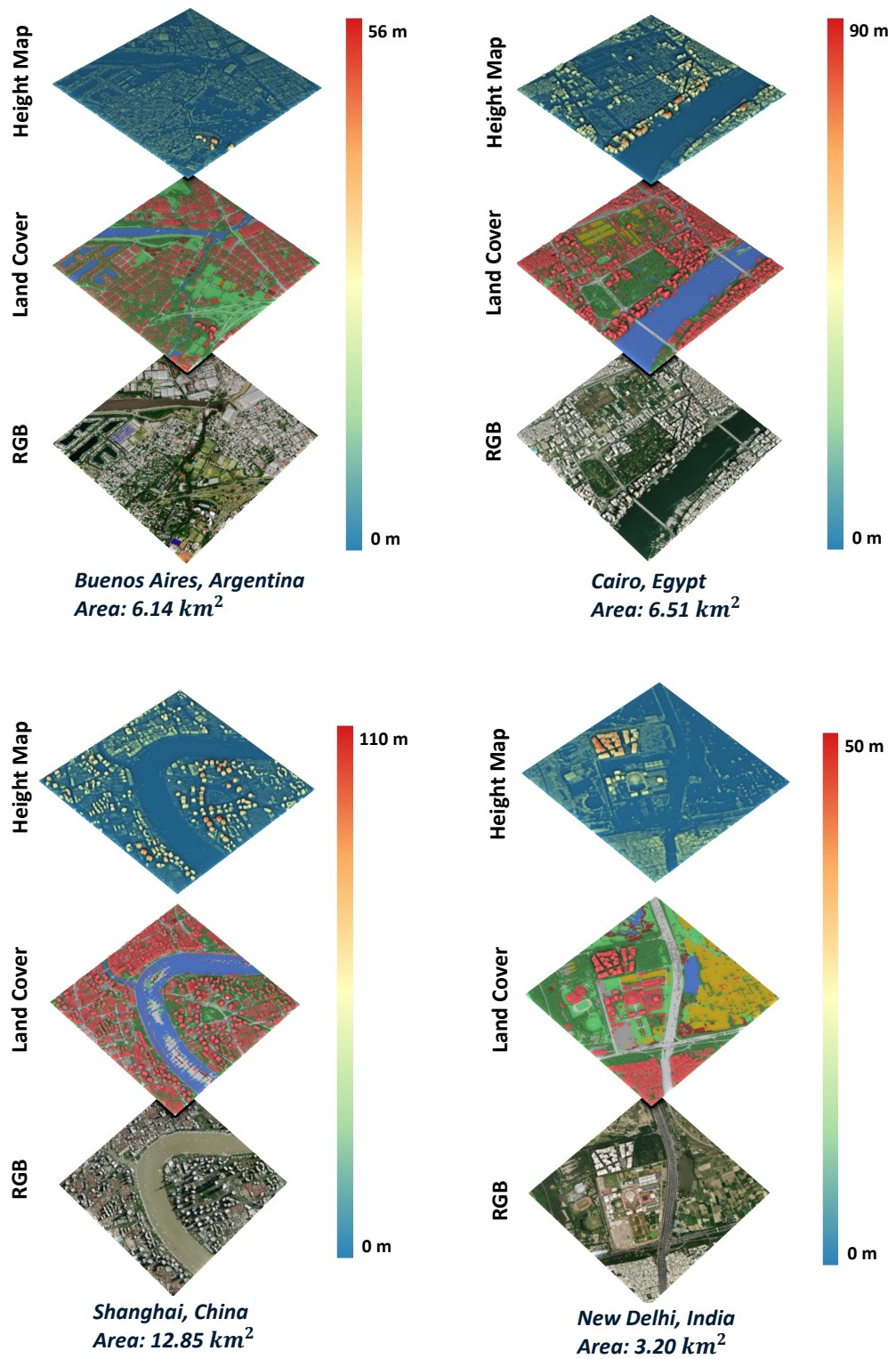

Figure 14: 3D visualization outcomes from real-world monocular RS images, which uses the model trained on SynRS3D dataset with proposed RS3DAda method. RGB satellite images of Buenos Aires and New Delhi: © HERE WeGo Satellite. RGB satellite images of Cairo and Shanghai: © Being Satellite.

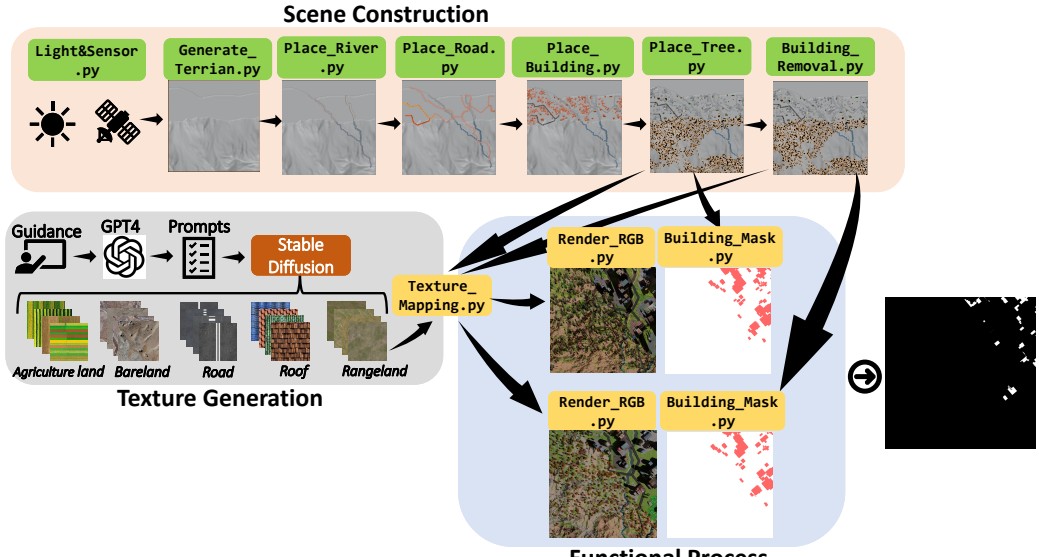

Figure 15: Generation workflow of building change detection mask.

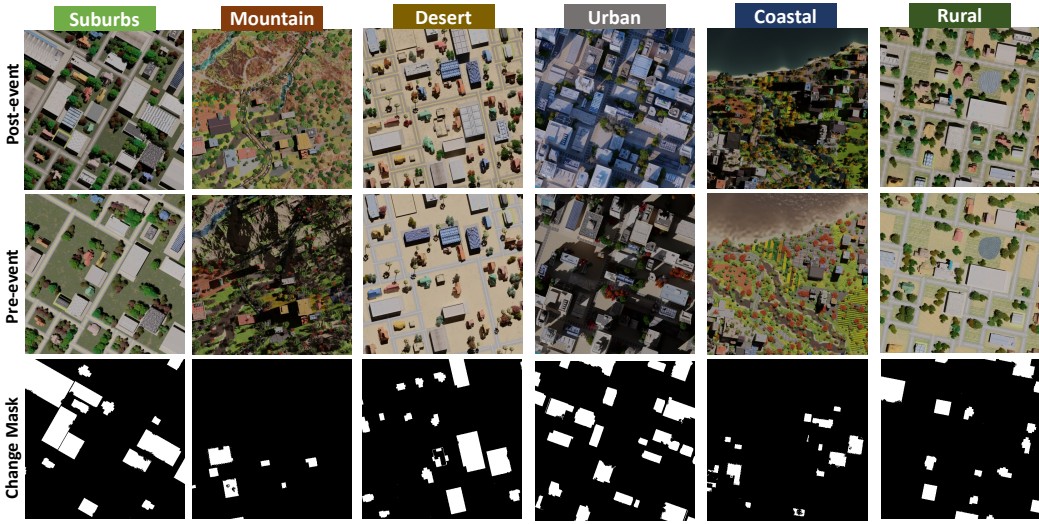

Figure 16: Examples of building change detection task in SynRS3D.

## A.8 Building Change Detection

SynRS3D provides 8-class land cover mapping annotations, accurate height maps, and binary masks specifically designed for RS building change detection tasks. The image and mask generation process is illustrated in Figure 15. For the synthesized scenes, an additional step is included: a certain proportion of buildings are randomly removed, and all geometries in the scene are retextured. Subsequently, post-event and pre-event RGB images, along with building masks, are rendered. By subtracting the two masks, the final building change mask is obtained. Figure 16 exhibits examples of change detection in six styles within SynRS3D.

To validate the effectiveness of SynRS3D in change detection tasks, we conducted experiments in a source-only scenario, where models were trained only on synthetic data and tested directly on real-world datasets. We compared our results with the models trained on two other advanced synthetic datasets, SMARS [76] and SyntheWorld [86], that include labels for the RS building change detection task. For real-world datasets, we used commonly utilized datasets in change detection tasks: WHU-CD [41], LEVIR-CD+ [9], and SECOND [108].

---

[3] https://www.bing.com/maps
[4] https://wego.here.com

The WHU-CD dataset, a subset of the WHU Building dataset, focuses on building change detection with aerial images from Christchurch, New Zealand, captured in April 2012 and 2016 at 0.3 meters/pixel resolution. Covering 20.5 km$^2$, the dataset documents significant urban development, with buildings increasing from 12,796 to 16,077 over four years. LEVIR-CD+ is an advanced building CD dataset comprising 985 pairs of high-resolution (0.5 meters/pixel) images, documenting changes over 5 to 14 years and featuring various building types. It includes 31,333 instances of building changes, making it a valuable benchmark for CD methodologies. The SECOND dataset consists of 4,662 pairs of 512×512 aerial images (0.5-3 meters/pixel) annotated for land cover change detection in cities like Hangzhou, Chengdu, and Shanghai, but in our experiments, we only use its building change mask. These datasets were split into training and testing sets in a 3:1 ratio, with a training size of 256×256 pixels.

We employed four change detection frameworks for evaluating SMAR, SyntheWorld, and SynRS3D, including the CNN-based DTCDSCN [59], the transformer-based ChangeFormer [6], and the current state-of-the-art Mamba-based method, ChangeMamba [10]. Notably, due to our empirical findings of the strong potential of DINOV2 [71] pre-trained networks on synthetic data in both land cover mapping and change detection tasks, we implemented a framework combining the DINOV2 encoder with the ChangeMamba decoder for change detection on synthetic data, which we named DINOMamba. For synthetic datasets, we use a batch size of 2, and for real data, we use a batch size of 16. The optimizer used is AdamW, with a learning rate of 1e-5 for DinoMamba, while all other methods use a learning rate of 1e-4. All models are trained for 40,000 iterations on a single Tesla A100. The evaluation metrics used are IoU and F1.

Table 12: Peformance evaluation of building change detection task on WHU-CD [41] dataset.

| Train on | DTCDSCN [59] | | ChangeFormer [6] | | ChangeMamba [10] | | DinoMamba | |
|---|---|---|---|---|---|---|---|---|
| | IoU | F1 | IoU | F1 | IoU | F1 | IoU | F1 |
| SMARS [76] | 26.84 | 42.55 | 18.67 | 31.88 | 42.50 | 59.63 | 48.11 | 64.87 |
| SyntheWorld [86] | 30.17 | 46.53 | **41.73** | **58.87** | 47.26 | 64.10 | 54.20 | 70.14 |
| SynRS3D | **33.09** | **49.84** | 35.00 | 51.94 | **52.94** | **69.08** | **61.60** | **76.00** |
| Real | 58.31 | 73.67 | 79.98 | 88.88 | 88.44 | 93.87 | 87.57 | 93.38 |

Table 13: Peformance evaluation of building change detection task on LEVIR-CD+ [9] dataset.

| Train on | DTCDSCN [59] | | ChangeFormer [6] | | ChangeMamba [10] | | DinoMamba | |
|---|---|---|---|---|---|---|---|---|
| | IoU | F1 | IoU | F1 | IoU | F1 | IoU | F1 |
| SMARS [76] | 11.70 | 21.53 | 15.67 | 27.58 | 27.50 | 42.50 | 30.85 | 47.31 |
| SyntheWorld [86] | 21.16 | 35.28 | 23.31 | 38.12 | 28.28 | 44.30 | 48.78 | 65.46 |
| SynRS3D | **25.82** | **41.30** | **23.33** | **38.14** | **30.39** | **46.78** | **49.63** | **66.23** |
| Real | 63.44 | 77.63 | 67.48 | 80.58 | 77.39 | 87.25 | 74.12 | 85.14 |

Table 14: Peformance evaluation of building change detection task on the SECOND [108] dataset.

| Train on | DTCDSCN [59] | | ChangeFormer [6] | | ChangeMamba [10] | | DinoMamba | |
|---|---|---|---|---|---|---|---|---|
| | IoU | F1 | IoU | F1 | IoU | F1 | IoU | F1 |
| SMARS [76] | 17.26 | 29.88 | 23.30 | 38.09 | 29.85 | 46.15 | 35.20 | 51.07 |
| SyntheWorld [86] | 21.00 | 35.07 | 26.44 | 42.06 | 27.23 | 43.02 | 37.61 | 54.71 |
| SynRS3D | **33.52** | **50.32** | **31.36** | **47.90** | **38.88** | **56.02** | **39.18** | **56.33** |
| Real | 58.78 | 74.04 | 60.08 | 75.06 | 67.61 | 80.68 | 67.65 | 80.71 |

Tables 12, 13, 14 present our experimental results, showing that the combination of SynRS3D and DINOMamba achieved F1 scores of 76.00, 66.23, and 56.33 on WHU, LEVIR-CD+, and SECOND respectively. Although there is still a gap compared to the Oracle model trained on real-world data, our dataset significantly boosts models' performances compared with the other two synthetic datasets. We have established a benchmark based on SynRS3D and advanced change detection networks, hoping to further promote the development of RS change detection using synthetic data.

## A.9 Disaster Mapping Study Cases

The models trained on the SynRS3D dataset using the RS3DAda method can be utilized for various remote sensing downstream applications. We explored their potential in disaster mapping applications.

In February 2023, a devastating earthquake struck southeastern Turkey, primarily affecting the Kahramanmaraş region. This earthquake, with a magnitude of 7.8, caused widespread destruction, resulting in over 45,000 deaths, thousands of injuries, and massive displacement of residents. The economic losses were estimated to be in

the billions of dollars. Rescue operations were carried out by both national and international teams, working tirelessly to save lives and provide aid to the affected population. Similarly, in August 2023, Hawaii experienced severe wildfires, particularly affecting the island of Maui. These wildfires, exacerbated by dry conditions and strong winds, led to extensive destruction of homes, infrastructure, and natural landscapes. The fires caused significant economic losses, displacing many residents and leading to casualties. The coordinated efforts of local authorities and fire departments, along with support from federal agencies, were crucial in controlling the fires and assisting those affected. To assess the impact of these disasters, we used the height estimation branch of RS3DAda to infer pre- and post-event remote sensing images. By simply subtracting the predicted height maps of the post-event from the pre-event, we obtained a Height Difference map. This map was filtered using a threshold: 3 meters for the earthquake example (indicating that buildings severely damaged in the earthquake would collapse, resulting in a significant height reduction) and 1 meter for the wildfire example (assuming that changes exceeding 1 meter indicate damage in the fire). Figure 18 presents the study case for the Turkey earthquake, and Figure 17 shows the study case for the Hawaii wildfires.

This simple method allowed us to roughly delineate the affected areas and assess the damage severity based on height differences. Although not entirely precise, this approach represents a significant success in applying models trained solely on synthetic data to real-world scenarios. We believe in the potential of RS3DAda and SynRS3D in this research domain and look forward to more applications and studies in the future.

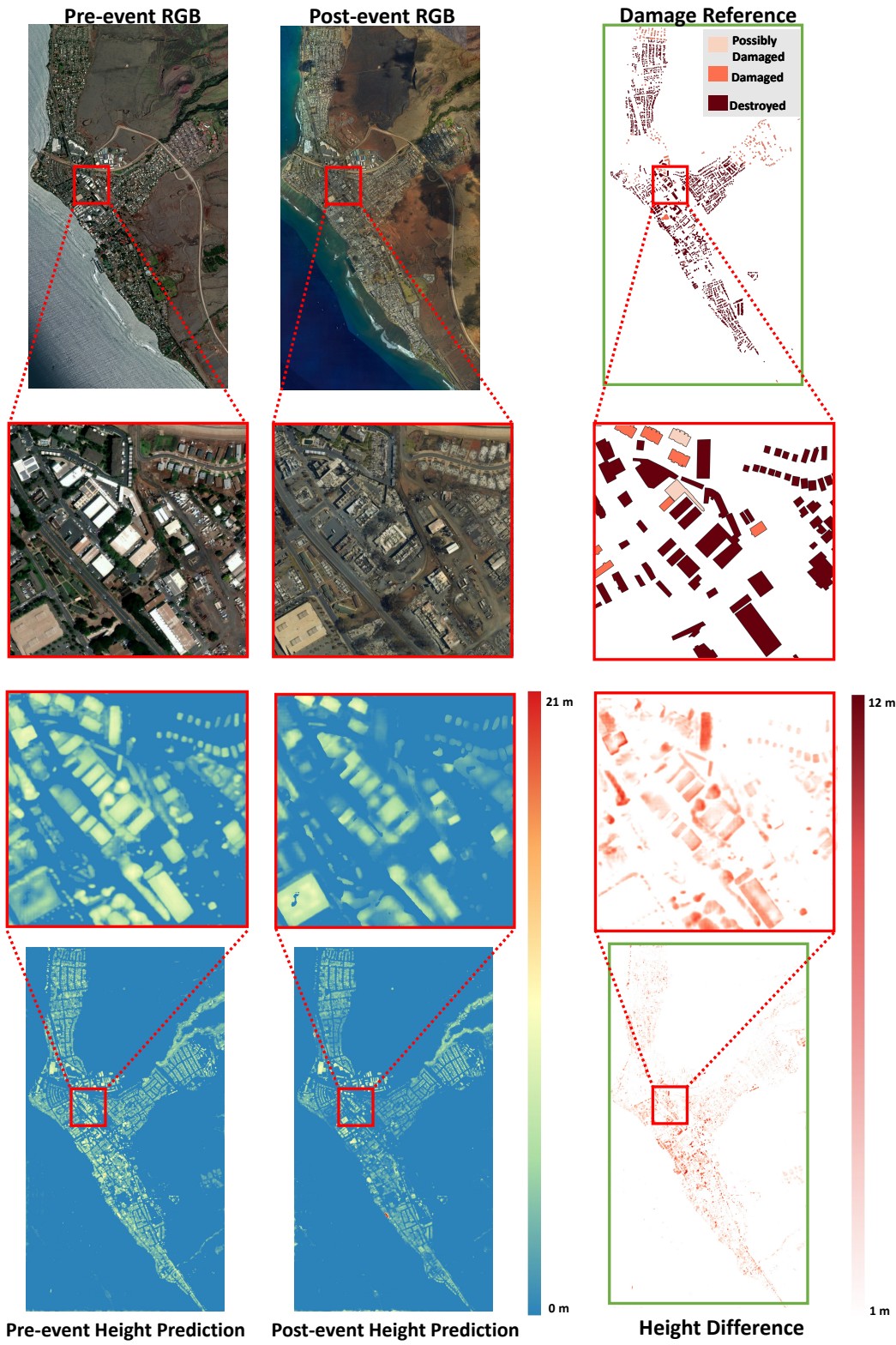

Figure 17: Study case of 2023 Hawaii-Maui wildfire. RGB satellite images of pre-event: © Being Satellite. RGB satellite images of post-event: © Google Satellite.

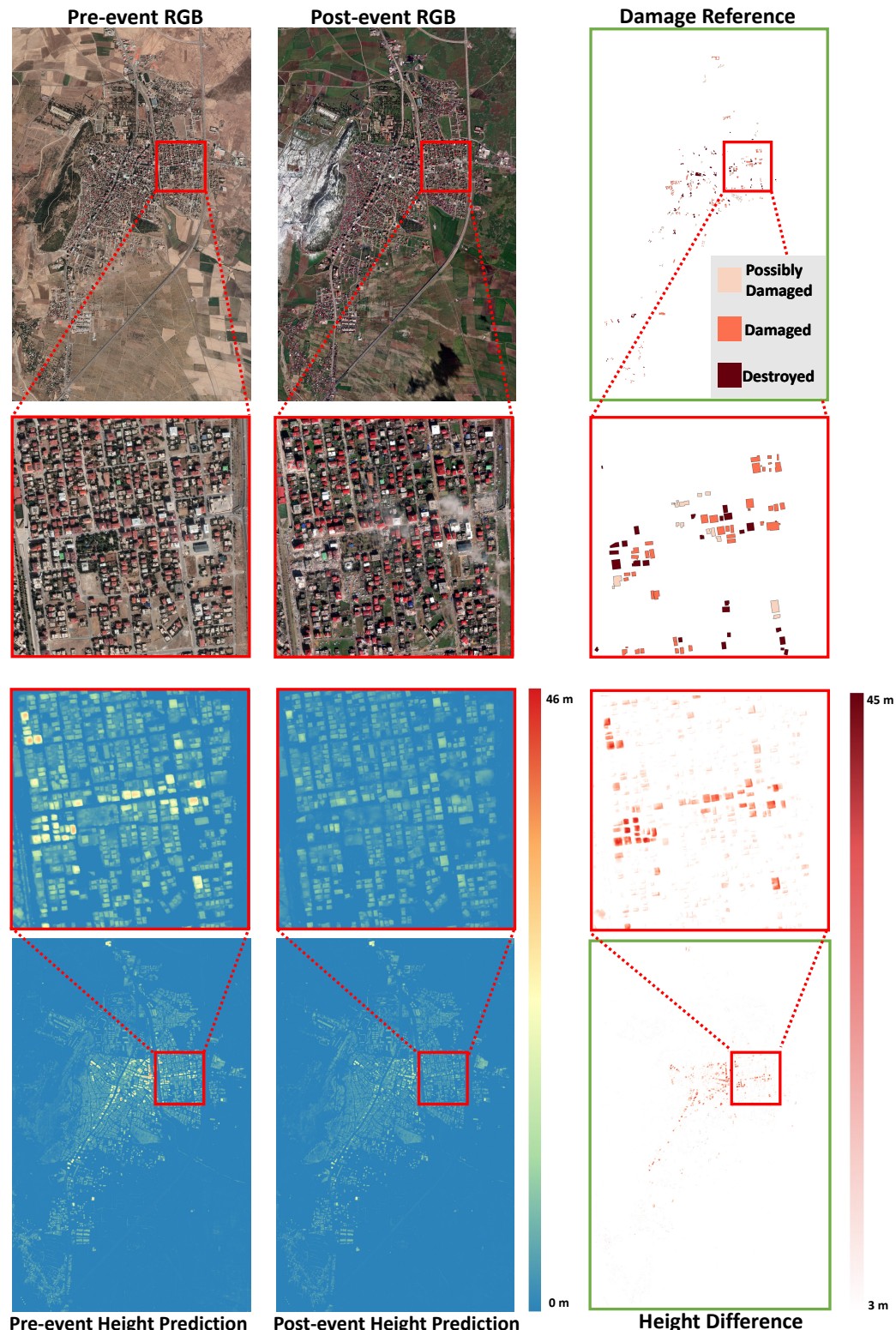

Figure 18: Study case of 2023 Turkey–Syria earthquakes. RGB satellite images: © 2023 CNES/Airbus, Maxar Technologies.

