# B  Datasheet

## B.1  Motivation

**1. For what purpose was the dataset created? Was there a specific task in mind? Was there a specific gap that needed to be filled? Please provide a description.**

**A1:** The dataset was created to enable global 3D semantic understanding from single-view high-resolution remote sensing imagery, addressing the challenges of high annotation costs, data collection, and geographically restricted data availability.

**2. Who created the dataset (e.g., which team, research group) and on behalf of which entity (e.g., company, institution, organization)?**

**A2:** SynRS3D is created by the first author and its affiliations.

**3. Who funded the creation of the dataset? If there is an associated grant, please provide the name of the grantor and the grant name and number.**

**A3:** This work was supported in part by JST FOREST Program Grant Number JPMJFR206S; Microsoft Research Asia; JSPS KAKENHI Grant Number 24KJ0652; the Next Generation AI Research Center of The University of Tokyo; the Japan Science and Technology Agency SPRING Program (JST SPRING) Grant Number JPMJSP2108; and RIKEN Junior Research Associate (JRA) Program.

## B.2  Composition

**1. What do the instances that comprise the dataset represent (e.g., documents, photos, people, countries)? Are there multiple types of instances (e.g., movies, users, and ratings; people and interactions between them; nodes and edges)? Please provide a description.**

**A1:** The instances are high-resolution synthetic RGB images representing diverse geographic environments and associated annotations for height estimation, land cover mapping, and building change detection. Land cover has 8 types: bareland, rangeland, developed space, road, tree, water, agricultural land, and building. The height range is from 0m to 409m.

**2. How many instances are there in total (of each type, if appropriate)?**

**A2:** 69,667 high-resolution RGB images. Specifically, it includes 69,667 pre-event images, 69,667 post-event images, corresponding height maps, land cover annotations, and building change detection masks.

**3. Does the dataset contain all possible instances or is it a sample (not necessarily random) of instances from a larger set? If the dataset is a sample, then what is the larger set? Is the sample representative of the larger set (e.g., geographic coverage)? If so, please describe how this representativeness was validated/verified. If it is not representative of the larger set, please describe why not (e.g., to cover a more diverse range of instances, because instances were withheld or unavailable).**

**A3:** SynRS3d aims to be representative by covering six different city styles worldwide. After the initial SynRS3D dataset generation, images with anomalous height distributions are filtered out. The final version of SynRS3D was constructed using the following prior knowledge [21]: backward regions (low buildings) cover about 12% of the world's areas, emerging regions (mid buildings) cover about 70%, and developed regions (tall buildings) cover about 18%. This step ensures that the final version of SynRS3D closely aligns with real-world height distributions. The specific filtering algorithm detail can be found in Algorithm 1.

**4. What data does each instance consist of? "Raw" data (e.g., unprocessed text or images) or features? In either case, please provide a description.**

**A4:** Each instance consists of high-resolution synthetic RGB images captured before and after the event, land cover labels, accurate height maps, and building change masks. The corresponding instances are organized into folders: `gt_cd_mask`, `gt_nDSM`, `gt_ss_mask`, `opt`, and `pre_opt`. Each folder contains the respective data, and each corresponding image within these folders shares the same name.

**5. Is there a label or target associated with each instance? If so, please provide a description.**

**A5:** Yes, labels include land cover mapping, height map, and building change mask. Land cover has 8 types: bareland, rangeland, developed space, road, tree, water, agricultural land, and building. The height range is from 0m to 409m.

**6. Is any information missing from individual instances? If so, please provide a description, explaining why this information is missing (e.g., because it was unavailable). This does not include intentionally removed information, but might include, e.g., redacted text.**

**A6:** Yes, while the current SynRS3D dataset already contains sufficient information, expanding each instance to include simulated SAR images would allow the dataset to be extended to various remote sensing multimodal tasks. However, successful simulation of SAR images is currently beyond our capabilities. This will be our future work.

**7. Are relationships between individual instances made explicit (e.g., users' movie ratings, social network links)? If so, please describe how these relationships are made explicit.**

**A7:** Our dataset does not include direct relationships between individual instances, and each instance is named numerically. However, the dataset is organized into multiple batches or packages, and the relationships between these batches can be inferred from their naming conventions. Each package is named using the format "xx_yy_zz", where "xx" represents the different layouts, "yy" denotes the ground sampling distance (GSD), and "zz" indicates whether the height distribution is low, medium, or high.

**8. Are there recommended data splits (e.g., training, development/validation, testing)? If so, please provide a description of these splits, explaining the rationale behind them.**

**A8:** As a synthetic dataset, the primary concern is evaluating the performance of models trained on this dataset when applied to real datasets. Therefore, we did not create training and testing splits for SynRS3D.

**9. Are there any errors, sources of noise, or redundancies in the dataset? If so, please provide a description.**

**A9:** No.

**10. Is the dataset self-contained, or does it link to or otherwise rely on external resources (e.g., websites, tweets, other datasets)? If it links to or relies on external resources, a) are there guarantees that they will exist, and remain constant, over time; b) are there official archival versions of the complete dataset (i.e., including the external resources as they existed at the time the dataset was created); c) are there any restrictions (e.g., licenses, fees) associated with any of the external resources that might apply to a dataset consumer? Please provide descriptions of all external resources and any restrictions associated with them, as well as links or other access points, as appropriate.**

**A10:** The dataset is self-contained.

**11. Does the dataset contain data that might be considered confidential (e.g., data that is protected by legal privilege or by doctor–patient confidentiality, data that includes the content of individuals' non-public communications)? If so, please provide a description.**

**A11:** No.

**12. Does the dataset contain data that, if viewed directly, might be offensive, insulting, threatening, or might otherwise cause anxiety? If so, please describe why.**

**A12:** No.

**13. Does the dataset identify any subpopulations (e.g., by age, gender)? If so, please describe how these subpopulations are identified and provide a description of their respective distributions within the dataset.**

**A13:** No.

**14. Is it possible to identify individuals (i.e., one or more natural persons), either directly or indirectly (i.e., in combination with other data) from the dataset? If so, please describe how.**

**A14:** No.

**15. Does the dataset contain data that might be considered sensitive in any way (e.g., data that reveals race or ethnic origins, sexual orientations, religious beliefs, political opinions or union memberships, or locations; financial or health data; biometric or genetic data; forms of government identification, such as social security numbers; criminal history)? If so, please provide a description.**

**A15:** No.

### B.3 Collection Process

**1. How was the data associated with each instance acquired? Was the data directly observable (e.g., raw text, movie ratings), reported by subjects (e.g., survey responses), or indirectly inferred/derived from other data (e.g., part-of-speech tags, model-based guesses for age or language)? If the data was reported by subjects or indirectly inferred/derived from other data, was the data validated/verified? If so, please describe how.**

**A1:** SynRS3D was generated using procedural modeling techniques based on Blender[4] and Python[5]. The RGB images were obtained using Blender's built-in orthographic camera and Cycles rendering engine. All distances and position information within the 3D software are known, so the labels were accurately obtained using Blender's compositor node. All images and labels are saved in TIFF format. The RGB images are three-channel uint8 format and can be viewed using various interactive image viewers (e.g., Windows Photos[6] or IrfanView[7]) or Python. Land cover annotations are single-channel uint8 format with values ranging from 1 to 8. Below is the label map and the colormap used for visualization in this work. Height map annotations are single-channel float32 format, and we recommend using more professional software such as QGIS[8] for visualization. The building change masks are single-channel images consisting of values 0 and 255.

```
colormap = {
    1: [181, 76, 76],       % Bareland
    2: [128, 255, 144],     % Grass
    3: [200, 200, 200],     % Developed space
    4: [242, 242, 242],     % Road
    5: [85, 160, 89],       % Tree
    6: [102, 153, 255],     % Water
    7: [246, 211, 45],      % Agriculture land
    8: [255, 102, 102]      % Buildings
}
```

**2. What mechanisms or procedures were used to collect the data (e.g., hardware apparatuses or sensors, manual human curation, software programs, software APIs)? How were these mechanisms or procedures validated?**

**A2:** Please refer to Section A.1.

**3. If the dataset is a sample from a larger set, what was the sampling strategy (e.g., deterministic, probabilistic with specific sampling probabilities)?**

**A3:** Please refer to Section A.1.

**4. Who was involved in the data collection process (e.g., students, crowdworkers, contractors) and how were they compensated (e.g., how much were crowdworkers paid)?**

**A4:** The first author of this paper.

---

[4] https://www.blender.org/

[5] https://www.python.org/

[6] https://support.microsoft.com/en-us/help/4026249/windows-10-photos

[7] https://www.irfanview.com/

[8] https://qgis.org/en/site/

**5. Over what timeframe was the data collected? Does this timeframe match the creation timeframe of the data associated with the instances (e.g., recent crawl of old news articles)? If not, please describe the timeframe in which the data associated with the instances was created.**

**A5:** Each instance took an average of 5 minutes to generate using a Tesla A100 GPU. We used 30-40 GPUs in parallel, and the total time taken was approximately one week.

### B.4 Preprocessing/Cleaning/Labeling

**1. Was any preprocessing/cleaning/labeling of the data done (e.g., discretization or bucketing, tokenization, part-of-speech tagging, SIFT feature extraction, removal of instances, processing of missing values)? If so, please provide a description. If not, you may skip the remaining questions in this section.**

**A1:** Yes, the data underwent a filtering process to remove images with anomalous height distributions.

**2. Was the "raw" data saved in addition to the preprocessed/cleaned/labeled data (e.g., to support unanticipated future uses)? If so, please provide a link or other access point to the "raw" data.**

**A2:** No.

**3. Is the software that was used to preprocess/clean/label the data available? If so, please provide a link or other access point.**

**A3:** No.

### B.5 Uses

**1. Has the dataset been used for any tasks already? If so, please provide a description.**

**A1:** No.

**2. Is there a repository that links to any or all papers or systems that use the dataset? If so, please provide a link or other access point.**

**A2:** N/A.

**3. What (other) tasks could the dataset be used for?**

**A3:** It can also be used for 3D change detection and disaster mapping applications.

**4. Is there anything about the composition of the dataset or the way it was collected and preprocessed/cleaned/labeled that might impact future uses? For example, is there anything that a dataset consumer might need to know to avoid uses that could result in unfair treatment of individuals or groups (e.g., stereotyping, quality of service issues) or other risks or harms (e.g., legal risks, financial harms)? If so, please provide a description. Is there anything a dataset consumer could do to mitigate these risks or harms?**

**A4:** No.

**5. Are there tasks for which the dataset should not be used? If so, please provide a description.**

**A5:** No.

### B.6 Distribution

**1. Will the dataset be distributed to third parties outside of the entity (e.g., company, institution, organization) on behalf of which the dataset was created? If so, please provide a description.**

**A1:** Yes, SynRS3D and related codes will be made publicly available.

**2. How will the dataset be distributed (e.g., tarball on website, API, GitHub)? Does the dataset have a digital object identifier (DOI)?**

**A2:** We provide Zenodo download link for SynRS3D.

**3. When will the dataset be distributed?**

**A3:** Now we provide Zenodo download link for SynRS3D.

**4. Will the dataset be distributed under a copyright or other intellectual property (IP) license, and/or under applicable terms of use (ToU)? If so, please describe this license and/or ToU, and provide a link or other access point to, or otherwise reproduce, any relevant licensing terms or ToU, as well as any fees associated with these restrictions.**

**A4:** It will be distributed under the Creative Commons Attribution-NonCommercial 4.0 International (CC BY-NC 4.0) License.

**5. Have any third parties imposed IP-based or other restrictions on the data associated with the instances? If so, please describe these restrictions, and provide a link or other access point to, or otherwise reproduce, any relevant licensing terms, as well as any fees associated with these restrictions.**

**A5:** No.

**6. Do any export controls or other regulatory restrictions apply to the dataset or to individual instances? If so, please describe these restrictions, and provide a link or other access point to, or otherwise reproduce, any supporting documentation.**

**A6:** No.

### B.7    Maintenance

**1. Who will be supporting/hosting/maintaining the dataset?**

**A1:** The authors.

**2. How can the owner/curator/manager of the dataset be contacted (e.g., email address)?**

**A2:** They can be contacted via email available on the GitHub repository.

**3. Is there an erratum? If so, please provide a link or other access point.**

**A3:** No.

**4. Will the dataset be updated (e.g., to correct labeling errors, add new instances, delete instances)? If so, please describe how often, by whom, and how updates will be communicated to dataset consumers (e.g., mailing list, GitHub)?**

**A4:** Yes, the authors will periodically review issues on GitHub and update the dataset based on the feedback.

**5. If the dataset relates to people, are there applicable limits on the retention of the data associated with the instances (e.g., were the individuals in question told that their data would be retained for a fixed period of time and then deleted)? If so, please describe these limits and explain how they will be enforced.**

**A5:** Not applicable.

**6. Will older versions of the dataset continue to be supported/hosted/maintained? If so, please describe how. If not, please describe how its obsolescence will be communicated to dataset consumers.**

**A6:** N/A.

**7. If others want to extend/augment/build on/contribute to the dataset, is there a mechanism for them to do so? If so, please provide a description. Will these contributions be validated/verified? If so, please describe how. If not, why not? Is there a process for communicating/distributing these contributions to dataset consumers? If so, please provide a description.**

**A7:** We have described the dataset generation process and all the tools used in detail in Section A.1. However, due to redistribution restrictions of the commercial add-ons used, we cannot provide the source code for synthetic data generation system. We are happy to assist anyone who wants to create or extend the dataset. Please contact the authors via email on our GitHub repository.