# OpenReview forum: "SynRS3D: A Synthetic Dataset for Global 3D Semantic Understanding from Monocular Remote Sensing Imagery"
_NeurIPS.cc/2024/Datasets_and_Benchmarks_Track — NeurIPS 2024 Track Datasets and Benchmarks Spotlight_

### Official Review · Reviewer_jAqv · 2024-07-24
**A meaningful benchmark**

**Rating:** 6
**Confidence:** 3
**Correctness:** Yes
**Clarity:** Yes

**Review:**

The paper presents a well-structured dataset and a novel UDA method for RS applications. The clarity of the writing is good, with a comprehensive explanation of the dataset generation process and the RS3DAda method. The originality is high, given the dataset's scale and diversity in RS.

**Pros**:

The introduction of SynRS3D and RS3DAda addresses major challenges in 3D semantic understanding from RS imagery. SynRS3D is the largest synthetic RS dataset, with extensive geographic and semantic coverage. RS3DAda provides a novel approach to domain adaptation, improving performance on real-world RS tasks. Extensive experiments validate the effectiveness and adaptability of the dataset and method. Clear explanation of the dataset generation and UDA method enhances reproducibility.

**Cons**:

Domain Gap: Despite RS3DAda, there remains a noticeable performance gap between synthetic and real-world data.

Specific Application: The UDA method is tailored to RS, potentially limiting its application to other fields.

Initial Overfitting: Early training instability on SynRS3D indicates a need for further refinement of the UDA method.

**Strengths:**

Contribution: SynRS3D and RS3DAda represent some advancements in the field of RS, addressing critical limitations in data availability and quality for 3D semantic understanding tasks.

Relevance to Broader Community: The work is highly relevant to the RS and broader machine learning communities, providing a valuable resource and methodology that can be applied to various environmental monitoring, urban planning, and disaster response applications.

**Additional Feedback:**

None

**Documentation:**

Still need the open source the dataset and its related workd such as data collection.

**Limitations:**

Yes

**Opportunities For Improvement:**

Please refer to the 'Cons' of the review

**Relation To Prior Work:**

Yes

**Summary And Contributions:**

The submission presents SynRS3D, a novel synthetic dataset aimed at enhancing global 3D semantic understanding from monocular remote sensing (RS) imagery. SynRS3D includes 69,667 high-resolution optical images covering various global city styles and land cover types, providing valuable annotations for land cover mapping and height estimation. Additionally, the paper introduces RS3DAda, a multi-task unsupervised domain adaptation (UDA) method designed to bridge the gap between synthetic and real-world RS data. The extensive experiments demonstrate the effectiveness and adaptability of SynRS3D and RS3DAda in real-world scenarios.

---

> ### Author Rebuttal · Authors · 2024-08-17
>
> Thank you for recognizing the importance of our research problem, the significance and novelty of our proposed datasets and methods, particularly in remote sensing, as well as the thoroughness of our experiments and providing information about reproduction! We hope the following responses can address your concerns and meet your expectations.
>
> > **Q1:** Documentation: Still need the open source the dataset and its related workd such as data collection.
>
> **A1:** Thank you for bringing attention to the placement of the SynRS3D dataset and RS3DAda code links. **These links can be found in lines 5-7 of the Appendix**, but we recognize they may have been easily overlooked. To improve accessibility, we will ensure these links are prominently included in the abstract in our revised submission.
>
> Regarding your concerns about data collection, we have enhanced the README file in our anonymous GitHub repository to provide comprehensive instructions. These include detailed steps for generating the SynRS3D dataset using Blender and Python, as well as guidelines for collecting and processing the real-world datasets utilized in our work. These resources are available under the directories `synthetic_data_generation` and `prepare_validation_datasets`.
>
> Please note that the SynRS3D dataset generation process involves the use of certain commercial Blender plugins. As we do not have the rights to redistribute these plugins, researchers interested in replicating our work will need to purchase them independently and follow the scripts we have provided. Additionally, some real-world validation datasets are subject to licensing restrictions; however, we have included detailed documentation and code to assist in their use, ensuring that others can effectively reproduce our results.
>
> We kindly ask that you review the provided documentation and reconsider the assessment of our reproducibility.
>
> > **Q2:** Domain Gap: Despite RS3DAda, there remains a noticeable performance gap between synthetic and real-world data.
>
> **A2:** Thank you for your insightful feedback. We recognize that there is still a performance gap between synthetic and real-world data, even with the introduction of RS3DAda. However, we would like to clarify several important points:
>
> 1. **Superior Performance in Source-Only Setting Compared to Existing Synthetic Datasets:**
>    Our dataset outperforms all existing synthetic remote sensing datasets in the source-only setting, indicating that the gap between SynRS3D and real-world data is smaller. When considering the broader field of computer vision semantic segmentation tasks, it is evident that significant gaps exist between synthetic and real-world performance in well-known datasets. For instance, the SYNTHIA [1] and GTA [2] datasets in autonomous driving exhibit gaps of 36 and 35 points in mIoU, respectively (Table 1 and Table 2)[3]. In the domain of point cloud segmentation, the Synth4D [4] and SynLiDAR [5] datasets show gaps of 29 and 25 points in mIoU, respectively. For UAV synthetic datasets like SKYSCENES [6] and SKYDRONE [7], the gaps are 38 and 46 points in mIoU, respectively. In comparison, the gap between SynRS3D and real-world data is 28 points in mIoU.
>
> 2. **Purpose of RS3DAda:**
>    We would like to clarify that the primary objective of RS3DAda is to bridge the gap by proposing the first framework specifically designed for UDA from synthetic to real domains in the context of multi-task dense prediction in remote sensing (Lines 101-102). In computer vision, existing UDA methods have reduced the accuracy gap between a model trained on the synthetic dataset GTA and a model trained on the real-world dataset Cityscapes [8] to approximately 10 points in mIoU. However, it is important to note that the **Cityscapes dataset mainly includes data from 50 European cities, most of which are in Germany**. In contrast, the real-world dataset, **OEM [9], used for evaluating RS3DAda and SynRS3D, covers 77 regions across 44 countries on six continents**, making it a highly challenging semantic segmentation dataset. From Table 5 of main text, RS3DAda improved SynRS3D’s performance from 40 to 48 points in mIoU, reducing the gap to 20 points compared to the real-world dataset's performance of 68 points. We believe this is a **respectable result for a baseline method**. Additionally, from Table 4 of main text, for height estimation tasks, we validated our model on 11 diverse datasets from various countries, achieving superior performance in challenging regions compared to models trained solely on real-world data from Western countries.
>
> 3. **Domain-Specific Design:**
>    From Table 5 and Table 6 of the main text, you can observe that existing computer vision UDA methods such as DAFormer [10], AdaptSeg [11], and DADA [12] do not perform as well in the RS domain as RS3DAda does. RS3DAda is specifically designed for remote sensing, incorporating RS image characteristics as priors. Moreover, RS3DAda is the first framework for synthetic-to-real multi-task UDA in RS, making it a highly relevant and meaningful baseline in the RS domain.
>
> We hope this clarifies our position and provides a stronger foundation for evaluating the significance and effectiveness of RS3DAda.

---

> > ### Author Rebuttal · Authors · 2024-08-17
> >
> > > **Q3:** Specific Application: The UDA method is tailored to RS, potentially limiting its application to other fields.
> >
> > **A3:** We appreciate the reviewers' insights regarding the specificity of the RS3DAda method to the RS domain. While it is true that RS3DAda and the SynRS3D dataset were crafted with the unique characteristics of RS data in mind, we believe that the underlying principles of RS3DAda hold potential for broader applications across different fields. For example, in street view imagery, the 'sky' class can parallel the 'ground' class in RS, enabling similar refinement in depth estimation tasks. We hope this explanation clarifies the broader applicability of our approach and illustrates its potential to inspire innovative methods beyond remote sensing.
> >
> >
> > > **Q4:** Initial Overfitting: Early training instability on SynRS3D indicates a need for further refinement of the UDA method.
> >
> > **A4:** As depicted in Figure 8 of the main manuscript, the green curve reflects the accuracy with RS3DAda applied, while the orange curve represents the Source-only accuracy. Before the integration of RS3DAda, the training process exhibited significant instability. However, the introduction of RS3DAda has substantially stabilized the training process. To address initial overfitting when training on SynRS3D, we have already incorporated several techniques into RS3DAda, including warm-up strategies, the use of small batch    sizes, and the implementation of feature constraints. These measures have been effective in reducing the initial instability. While we acknowledge that some minor instability persists, we are committed to further refining our approach.
> >
> > We hope that this response clarifies that RS3DAda has already made significant strides in improving training stability, and we are dedicated to continuing our efforts to refine and enhance the method.
> >
> >
> > **References:**
> >
> > [1] Richter, Stephan R., et al. "Playing for data: Ground truth from computer games." Computer Vision–ECCV 2016: 14th European Conference, Amsterdam, The Netherlands, October 11-14, 2016, Proceedings, Part II 14. Springer International Publishing, 2016.
> >
> > [2] Ros, German, et al. "The synthia dataset: A large collection of synthetic images for semantic segmentation of urban scenes." Proceedings of the IEEE conference on computer vision and pattern recognition. 2016.
> >
> > [3] Xie, Binhui, et al. "Towards fewer annotations: Active learning via region impurity and prediction uncertainty for domain adaptive semantic segmentation." Proceedings of the IEEE/CVF conference on computer vision and pattern recognition. 2022.
> >
> > [4] Xiao, Aoran, et al. "Transfer learning from synthetic to real lidar point cloud for semantic segmentation." Proceedings of the AAAI conference on artificial intelligence. Vol. 36. No. 3. 2022.
> >
> > [5] Saltori, Cristiano, et al. "Gipso: Geometrically informed propagation for online adaptation in 3d lidar segmentation." European Conference on Computer Vision. Cham: Springer Nature Switzerland, 2022.
> >
> > [6] Khose, Sahil, et al. "SkyScenes: A Synthetic Dataset for Aerial Scene Understanding." arXiv preprint arXiv:2312.06719 (2023).
> >
> > [7] Rizzoli, Giulia, et al. "Syndrone-multi-modal uav dataset for urban scenarios." Proceedings of the IEEE/CVF International Conference on Computer Vision. 2023.
> >
> > [8] Cordts, Marius, et al. "The cityscapes dataset for semantic urban scene understanding." Proceedings of the IEEE conference on computer vision and pattern recognition. 2016.
> >
> > [9] Xia, Junshi, et al. "Openearthmap: A benchmark dataset for global high-resolution land cover mapping." Proceedings of the IEEE/CVF Winter Conference on Applications of Computer Vision. 2023.
> >
> > [10] Hoyer, Lukas, Dengxin Dai, and Luc Van Gool. "Daformer: Improving network architectures and training strategies for domain-adaptive semantic segmentation." Proceedings of the IEEE/CVF conference on computer vision and pattern recognition. 2022.
> >
> > [11] Tsai, Yi-Hsuan, et al. "Learning to adapt structured output space for semantic segmentation." Proceedings of the IEEE conference on computer vision and pattern recognition. 2018.
> >
> > [12] Vu, Tuan-Hung, et al. "Dada: Depth-aware domain adaptation in semantic segmentation." Proceedings of the IEEE/CVF International Conference on Computer Vision. 2019.

---

> > > ### Comment · Reviewer_jAqv · 2024-08-26
> > >
> > > I appreciate the authors' responses. I've read the other reviews and the authors' replies, and my concerns have been addressed. Therefore, I will raise my score.

---

> > > > ### Author Response · Authors · 2024-08-26
> > > >
> > > > Thank you once again for your valuable feedback and suggestions. We are pleased to have had the opportunity to address your concerns. If you have any further questions or require additional clarification, please feel free to let us know.

---

### Official Review · Reviewer_7f97 · 2024-07-24

**Rating:** 7
**Confidence:** 4
**Correctness:** Yes
**Clarity:** Yes

**Review:**

The paper presents a significant contribution to Earth Observation by creating SynRS3D, the largest synthetic remote sensing (RS) 3D dataset, and developing the RS3DAda method for unsupervised domain adaptation. Here's an evaluation of the paper:

Pros:
1. Comprehensive dataset with 69,667 high-resolution optical images.
2. Novel method proposed (RS3DAda).
3. Extensive experimental validation on real-world datasets.
4. The dataset and codebase are well-documented and have a broad impact on various RS applications.
5. Extensive benchmarking experiments and analysis are provided.

Cons:
1. The generated images are not realistic. Synthetic data may not fully capture real-world complexities.
2. Scalability to diverse regions requires further investigation.
3. Need for validation in a wider range of real-world scenarios and tasks to strengthen significance.

**Strengths:**

The proposed dataset and benchmarking results are good sources for the community. This dataset has the potential to be used in many different tasks, including multi-modal foundation model training.

**Additional Feedback:**

N/A

**Documentation:**

Yes

**Opportunities For Improvement:**

The generated images are not realistic. Some unsupervised style transfer or unpaired Image-to-Image translation methods could be applied to make them more photo-realistic.

**Relation To Prior Work:**

Yes

**Summary And Contributions:**

The paper addresses the challenges of global semantic 3D understanding from single-view high-resolution remote sensing (RS) imagery, which is crucial for Earth Observation (EO). These challenges include high annotation costs, data collection expenses, and geographically restricted data availability. To overcome these issues, the authors introduce SynRS3D, the largest synthetic RS 3D dataset, created through a specialized synthetic data generation pipeline. SynRS3D consists of 69,667 high-resolution optical images featuring six different city styles, eight land cover types, precise height information, and building change masks. Additionally, the authors develop RS3DAda, a novel multi-task unsupervised domain adaptation (UDA) method, to facilitate the transition from synthetic to real scenarios for land cover mapping and height estimation. Extensive experiments on various real-world datasets demonstrate the effectiveness and adaptability of SynRS3D and RS3DAda, ultimately enabling global monocular 3D semantic understanding based on synthetic data. The SynRS3D dataset and related codes will be made available for further research.

---

> ### Author Rebuttal · Authors · 2024-08-17
>
> Thank you for acknowledging the merit of our data, the rigor of our experiments and analysis, and the novelty of our proposed method. We address your concerns in detail below.
>
> > **Q1:** The generated images are not realistic. Some unsupervised style transfer or unpaired Image-to-Image translation methods could be applied to make them more photo-realistic.
>
> **A1:** Thank you for your valuable suggestion! Based on your suggestion, we experimented with using three commonly used methods, FDA [1], HM [2], and PDA [3], for style transfer. To estimate the domain gap between the source domain after style transfer and the target domain (realistic images), we tested the source-only model's performance on the target domain after applying the style transfer. The height estimation results are shown in the table below:
>
> | Model                 | MAE ↓ (Whole) | MAE ↓ (High) | RMSE ↓ (Whole) | RMSE ↓ (High) | F1^HE ↑ (δ < 1.25) | F1^HE ↑ (δ < 1.25^2) | F1^HE ↑ (δ < 1.25^3) |
> |-----------------------|---------------|--------------|----------------|---------------|----------------------------------------|---------------------------------------------|----------------------------------------------|
> | **Avg. T.D.1**        |               |              |                |               |                                        |                                             |                                              |
> | Source Only           | 2.557         | **5.617**    | 4.128          | **6.705**     | 0.372                                  | 0.491                                       | 0.552                                        |
> | Source Only + FDA     | 2.603         | 5.822        | 4.321          | 7.087         | 0.342                                  | 0.476                                       | 0.549                                        |
> | Source Only + HM      | 2.435         | 5.842        | 4.225          | 7.057         | **0.398**                               | **0.524**                                   | **0.578**                                    |
> | Source Only + PDA     | **2.307**     | 5.858        | **4.095**      | 7.139         | 0.376                                  | 0.501                                       | 0.565                                        |
> | **Avg. T.D.2**        |               |              |                |               |                                        |                                             |                                              |
> | Source Only           | 6.117         | 8.923        | 9.221          | 11.443        | 0.365                                  | 0.514                                       | 0.601                                        |
> | Source Only + FDA     | 6.190         | 9.123        | 9.502          | 11.893        | 0.376                                  | 0.530                                       | 0.619                                        |
> | Source Only + HM      | 5.919         | **8.765**    | 9.147          | **11.413**    | **0.392**                               | **0.540**                                   | **0.624**                                    |
> | Source Only + PDA     | **5.785**     | 8.869        | **9.129**      | 11.628        | 0.375                                  | 0.520                                       | 0.597                                        |
>
>
> The results of land-cover mapping are as follows:
>
> | Model                 | Bareland | Rangeland | Developed | Road  | Tree  | Water | Agriculture | Buildings | **mIoU** |
> |-----------------------|----------|-----------|-----------|-------|-------|-------|-------------|-----------|----------|
> | Source-only           | **8.69** | **37.95**  | 22.54     | **49.05** | 60.16 | 46.64 | 35.40       | **65.19**  | 40.70    |
> | Source-only + FDA     | 3.25     | 36.24      | 18.35     | 47.54 | 57.89 | 41.68 | **57.49**   | 62.14     | 40.57    |
> | Source-only + HM      | 5.11     | 29.22      | **24.12** | 46.12 | 61.55 | **68.76** | 56.56       | 57.73     | **43.65** |
> | Source-only + PDA     | 8.06     | 26.84      | 21.91     | 48.35 | **63.05** | 67.59 | 55.92       | 50.96     | 42.84    |
>
> Based on your valuable suggestions, we observed an improvement in the performance of the source-only model after applying style transfer to the dataset. This demonstrates that style transfer methods effectively reduce the domain gap, making the SynRS3D data more realistic. Additionally, we provide visual examples of the images after style transfer; please refer to **Figure 1 in the PDF for a global response**. We will update the manuscript to showcase these results and also release the dataset after style transfer for further research.

---

> > ### Author Rebuttal · Authors · 2024-08-17
> >
> > > **Q2:** Scalability to diverse regions requires further investigation.
> >
> >
> > **A2:** Thank you for your insightful comment. Scalability is indeed crucial, and we have designed our dataset with this in mind, covering six distinct city styles from around the world. We have taken significant steps to validate the scalability of our dataset and the RS3DAda model across diverse regions.
> >
> > **Land-cover mapping:**  We tested the RS3DAda model, trained on SynRS3D, using the OEM [4] dataset, which encompasses 97 regions across 44 countries on six continents. The OEM dataset includes high-resolution images with detailed eight-class land cover annotations. Additionally, we evaluated the model on four other datasets—Vaihingen, Potsdam, JAX, and OMA—each derived from different sensors and representing diverse landscapes. These tests demonstrate the model's adaptability and scalability across various global environments.
> >
> > **Height estimation:**  For height estimation, the model was tested on 11 geographically distributed datasets, as detailed in Table 2 of Section 5. Although the diversity of regions for height estimation is currently more limited compared to land-cover mapping, this is due to the significant challenges in acquiring high-resolution elevation data globally. Nevertheless, we have made considerable efforts to validate the scalability of SynRS3D for this task.
> >
> > We acknowledge the importance of further investigating scalability and are committed to expanding our validation regions in future research to continue demonstrating the robustness of our dataset and approach.
> >
> >
> > > **Q3:** Need for validation in a wider range of real-world scenarios and tasks to strengthen significance.
> >
> > **A3:** Thank you for highlighting the importance of validating our work across diverse real-world scenarios. We have undertaken several additional experiments to demonstrate the broad applicability and significance of the SynRS3D dataset and the RS3DAda model. Beyond the height estimation and land cover mapping tasks detailed in the main manuscript, we have extended our validation to include:
> >
> > **Building Change Detection:** Detailed in Section A.8 of the Appendix, this task assesses the model's ability to identify building changes over time, which is crucial for urban planning .
> >
> > **Disaster Mapping:** As elaborated in Section A.9 of the Appendix, this application evaluates the model's effectiveness in identifying areas affected by natural disasters, aiding in rapid response and recovery efforts.
> >
> > These additional validations underscore the dataset's adaptability and relevance of the dataset to pressing real-world challenges.
> >
> > We further explored the potential of the RS3DAda model trained on SynRS3D in the context of **High-Resolution Canopy Height Mapping**. Accurate canopy height estimation is vital for understanding forest biomass, carbon storage, and ecological dynamics. We compared our model's performance with that of a large foundation model trained on an extensive real-world dataset, as referenced in [5], denote as HRCHM in the below table:
> >
> > | Dataset     | Model   | MAE ↓ | RMSE ↓ | F1^HE ↑ |
> > |-------------|---------|-------|--------|------------------------------|
> > | **JAX**     | HRCHM   | 1.295 | 3.379  | 0.620                        |
> > |             | RS3DAda | **1.226** | **2.586** | **0.653**              |
> > | **OMA**     | HRCHM   | 0.726 | 2.294  | 0.403                        |
> > |             | RS3DAda | **0.653** | **2.040** | **0.507**              |
> > | **Vaihingen** | HRCHM | **0.785** | 2.454  | 0.130                      |
> > |             | RS3DAda | 1.164 | **2.315** | **0.451**                 |
> > | **Potsdam** | HRCHM   | **0.507** | **2.264** | 0.013                    |
> > |             | RS3DAda | 0.682 | 2.349  | **0.297**                   |
> >
> > The results demonstrate that the RS3DAda model achieves highly competitive outcomes, closely matching those of models trained on vast amounts of real data, also we give some visualization results of this application in **Figure 2 of PDF file in Global Rebuttal**.
> >
> > To date, SynRS3D has showcased excellent performance across five critical remote sensing applications: **land cover mapping, height estimation, building change detection, disaster mapping, and high-resolution canopy height mapping**. We are committed to further exploring and validating its applicability in other practical domains. We appreciate the reviewer's valuable feedback, which has guided us in strengthening the breadth and depth of our evaluations.
> >
> > **References:**
> >
> > [1] Yang, Yanchao, and Stefano Soatto. "Fda: Fourier domain adaptation for semantic segmentation." Proceedings of the IEEE/CVF conference on computer vision and pattern recognition. 2020.
> >
> > [2] R. C. Gonzalez and R. E. Woods, Digital Image Processing (3rd Edition). Prentice-Hall, Inc., 2006
> >
> > [3] X. Xiao and L. Ma, “Color transfer in correlated color space,” in Proceedings of the 2006 ACM international conference on Virtual reality continuum and its applications, pp. 305–309, ACM, 2006.
> >
> > [4] Xia, Junshi, et al. "Openearthmap: A benchmark dataset for global high-resolution land cover mapping." Proceedings of the IEEE/CVF Winter Conference on Applications of Computer Vision. 2023.
> >
> > [5] Tolan, Jamie, et al. "Very high resolution canopy height maps from RGB imagery using self-supervised vision transformer and convolutional decoder trained on aerial lidar." Remote Sensing of Environment 300 (2024): 113888.

---

### Official Review · Reviewer_ezGZ · 2024-08-14
**SynRS3D Review**

**Rating:** 8
**Confidence:** 4

**Review:**

The SynRS3D dataset introduced is far from novel, as there are plenty of existing synthetic 3D building height estimation datasets out there, but the size of the dataset and careful attention to coverage of different types of cities on all 6 continents make it the new state-of-the-art dataset for supervised pre-training for monocular height estimation from remote sensing. As the authors mention in their introduction, this is an important task for a number of applications including building mapping, urban planning, and natural disaster response.

The RS3DAda domain adaptation method introduced is very interesting, and is far more novel than the dataset. In combination with the dataset, these contributions make a very strong contribution to the literature. This work is generally of high quality, with promising results on almost all evaluation benchmarks. The benchmarks are all standard, allowing for easy comparisons with other works. The experimental setup, hyperparameters used, and even versions of software used are all carefully documented, making reproducibility of no concern.

Pros:

* SynRS3D shows a lot of promise as a dataset for model pre-training, with great size and diversity of tasks
* RS3DAda shows a lot of promise for bridging the gap between synthetic and real data sources
* Benchmark experiments are very thorough and results look great

Cons:

* Related work could be made more clear, specifically how this dataset compares to existing non-synthetic datasets
* SynRS3D generation is a bit unclear without seeing the source code
* Almost too many figures/tables, feels like SynRS3D and RS3DAda deserve two separate papers

**Strengths:**

* Largest synthetic remote sensing dataset ever created, at least for 3D building height estimation (not clear on this)
* Global coverage of city morphologies, especially important for the Global South
* First use of multi-task UDA for synthetic-to-real domain adaptation in remote sensing
* Appendix is even longer than the main text, with tons of valuable experiments and results for the avid reader
* This includes ablation studies, extensions to building change detection, and applications in natural disaster response
* Most code will be made public, and installation instructions even include the specific versions of software used

**Additional Feedback:**

Minor comments:

* Introduction: The first paragraph may benefit from first introducing 3D reconstruction as a common task in computer vision before introducing EO data as a subdomain and 2D monocular height estimation as a specific task
* Related Work: I would use the \paragraph{} command for each subsection instead of just \textbf{}
* Table 1: An en dash (--) should be used for numerical ranges instead of a tilde (~) in the GSD column
* Line 187: "[100]" -> "Yang et al. [100]", see "citations as words": https://www.ece.ucdavis.edu/~jowens/commonerrors.html
* Table 2: Text is too small, make this bigger. I would also avoid using vertical lines when using the booktabs package
* Figure 6: DINO should be capitalized
* Table 4: 'Whole' and 'High' should use backticks for the first quotation mark in LaTeX
* References: Sorting references by order of appearance would allow citations like  [17, 79, 110, 37, 42, 44, 45, 56, 46, 21, 18, 41] to collapse to [5–17], saving space and giving you more room. Need to be careful with capitalization, methods like OpenEarthMap are displayed as Openearthmap. GPT-4 technical report is missing a list of authors.

Typos:

* Figure 2: "statisticals" -> "statistics"
* Line 212: "Senarios" -> "Scenarios"

Grammar:

* Line 61: "to advance" -> "can advance"
* Line 217: "examining" -> "demonstrating" or "illustrating"

**Clarity:**

The biggest gift and curse of this paper is the sheer volume of contents in the paper. The authors introduce a fantastic dataset, a novel domain adaptation strategy, a new evaluation metric, and tons of downstream applications, all of which could easily be their own paper. The downside of this is that some sections like Section 3 feel too short and not-detailed enough, while others like Section 5 feel too long and cluttered. There are so many tables and figures in Section 5 that I don't know where to start looking. I don't know if it's possible to improve too much on this within the constraints of a 9-page paper. The appendix is already extremely long (and interesting!), so it's not like moving more stuff to the appendix will help.

Specific comments:

* Abstract: the phrase "easily accessible" is unclear to me. Do you mean that synthetic data is easy to acquire/generate compared to conventional data?
* Figure 1: Caption needs more details to clarify that the top colorbar is for row 2 and the right colorbar is for row 1
* Line 47: Although UDA has been defined in the abstract, I would define this acronym one more time here
* Figure 2: Row 2 is missing a colorbar to explain the meaning of each color. I'm guessing it's the same as Figure 1, but each figure should be standalone. The caption needs much more detail to explain the subfigures. I'm unfamiliar with boxen plots, so the top right subfigure in particular isn't clear to me. Additionally, the 2 rightmost subfigures are far too small. I would suggest moving them to a separate figure, either in the main text or in the appendix.
* Figure 5: I'm not a huge fan of radar charts, I think a bar chart may be more easy to read here
* Section 5.1: It wasn't clear to me what the "source-only scenario" was at first until I reread previous sections of the paper. May be worth clarifying the experimental setup again here. I believe it's just "zero shot" prediction, but could be wrong
* Table 3: Missing units (IoU), could be added to caption

**Correctness:**

Use of a DPT model makes perfect sense, as the model was specifically designed with monocular depth estimation in mind.

Use of an L1 loss instead of the more common L2 loss makes sense for building height estimation, where most pixels should be 0. With an L2 loss, it's likely that the results would see more short buildings so the model can avoid large mistakes.

Main text:

* Line 41: what does 0 meter GSD mean? According to Table 1 it should actually be 0.09 meters
* Line 56: is it the largest synthetic RS dataset (Introduction) or the largest synthetic 3D RS dataset (Abstract)?

Appendix:

* Table 3: Why is the batch size so small?

**Documentation:**

Dataset construction is documented in Section 3.2, but not nearly in enough detail to reproduce the work. Code cannot be made public due to licensing restrictions. This is quite unfortunate, as this pipeline could be used for a lot of other applications or for increasing the size of the dataset in the future.

The authors document the train-test splits used, but not the exact images in each split. It would be good to either document the random seed used to split the data, or keep a list of images used in each split for reproducibility purposes.

I would like to commend the authors on documenting the exact versions of every single Blend add-on and many of the Python dependencies, this is fantastic. They even document the exact Albumentations implementations used for data augmentations, which are always different across different frameworks.

**Ethics:**

The great thing about synthetic data is that there aren't a lot of possible ethical issues or negative societal impacts.

**Limitations:**

The great thing about synthetic data is that there aren't a lot of possible ethical issues or negative societal impacts. The discussion section of this paper is clearly designed to satisfy the requirements of NeurIPS and please the reviewers, but this reviewer personally doesn't think it is necessary. I would personally be fine with removing this section entirely or moving it to the appendix. If anything, the discussion could instead focus on the limitations of the dataset (no dataset can possibly encompass the diversity of all global cities) and the uncertainty in the domain adaptation evaluations.

**Opportunities For Improvement:**

* Related work: See section below, would like to see more explanation of 2D monocular reconstruction and comparisons with non-synthetic datasets
* Figures: Many figures are too small to read and are very cluttered, feels like I'm reading a Nature paper (in both good and bad ways)
* Tables: Lacking error bars, making it hard to tell if the dataset/method are actually better or if it's random chance
* Captions: Many figure and table captions are insufficient to understand the display item. Each figure/table should be standalone without requiring the reader to see the main text to grasp the content
* Metrics: I don't know if I've seen sufficient evidence to demonstrate that the proposed F1^HE metric is better than existing metrics. This feels like one-too-many contributions for one paper. This metric has the potential for use for a wide range of regression tasks with long-tail distributions, not just height estimation. This metric deserves its own paper before it's really ready for widespread use

**Relation To Prior Work:**

The Related Works section primarily focuses on introducing what has already been done, but doesn't go as deeply into how this paper compares to those prior works. A more detailed discussion of Table 1 in this section would help.

I would rename "High-Resolution Earth Observation" and focus more specifically on 3D reconstruction in RS, with specific emphasis on monocular height estimation. The limitations of high-res EO should be familiar to almost anyone to which this paper might be useful, but readers may be less familiar with how monocular height estimation works.

Table 1 only includes synthetic RS datasets. A comparison with non-synthetic 3D building reconstruction datasets would improve the quality of the paper by highlighting the small size of non-synthetic datasets for this task. This would reinforce the idea that synthetic data is needed.

**Summary And Contributions:**

In this work, the authors introduce SynRS3D, a synthetic dataset for 3D semantic reconstruction (regression) and land cover mapping (semantic segmentation) from 2D monocular remote sensing imagery. They also introduce RS3DAda, a novel multi-task unsupervised domain adaptation (UDA) method, to improve the transferability of models trained on this synthetic data to real-world settings. They include a variety of benchmarks in zero-shot, few-shot, and transfer-learning scenarios to demonstrate the efficacy of models trained on their dataset.

Specific contributions include:

* SynRS3D: the largest synthetic remote sensing dataset
* RS3DAda: the first use of multi-task UDA for synthetic-to-real domain adaptation in remote sensing
* F1^HE: a new metric based on F1-score for regression of building height estimation
* A series of benchmarks validating the SynRS3D dataset and RS3DAda domain adaptation method

---

> ### Author Rebuttal · Authors · 2024-08-17
>
> Thank you very much for recognizing our work and for your meticulous suggestions. These recommendations will significantly enhance the quality of our paper! Below, we will address your concerns one by one:
>
> > **Q1:** Related work could be made more clear, specifically how this dataset compares to existing non-synthetic datasets.
>
> **A1:** Thank you for your valuable suggestion. In fact, in Table 2 of the main text, we listed some important attributes of all the non-synthetic datasets used. We acknowledge that providing a clearer comparison with existing non-synthetic datasets would enhance the clarity and significance of our work. While we compared 8 synthetic datasets and utilized 14 real-world datasets (11 for height estimation, 1 for semantic segmentation, and 3 for building change detection) in our experiments, we recognize that detailing all of these comparisons in the main text could overwhelm the reader with information.
>
> To address this, we will include a more comprehensive table in the appendix that compares both synthetic and non-synthetic datasets, providing a clearer context for our contributions. This table will highlight key characteristics, making it easier to understand how SynRS3D stands out in relation to existing datasets.
>
> > **Q2:** SynRS3D generation is a bit unclear without seeing the source code.
>
> **A2:** We completely understand your concern. Therefore, we have decided to release the code for the generation system.
>
> The updated section in our anonymous GitHub repository now includes the source code for SynRS3D generation. **Interested users will need to purchase these plugins independently**, but fortunately, they are not expensive. We will also provide updated documentation on our modifications to these plugins and the refactored generation code to ensure the community can reproduce our work and develop their data generation pipeline.
>
> > **Q3:** Tables: Lacking error bars, making it hard to tell if the dataset/method are actually better or if it's random chance.
>
> **A3:** Thank you for your feedback. We reported the average accuracy across five runs for the experimental results in Tables 4 and 5. Additionally, the shaded regions in Figure 8 represent error margins to some extent, illustrating that while training under source-only or zero-shot settings can be quite unstable, the RS3DAda method helps stabilize the training process.
>
> To clarify that our superiority is not due to random chance, we have added the standard deviations for the two most important experiments (Tables 4 and 5). The experimental results are as follows, where HE represents the height estimation task and LC represents the land cover mapping task. Since DAFormer is specifically designed for semantic segmentation/land cover mapping, we did not report its results in the height estimation task:
>
> | Task | Model            | MAE    ↓       | Task | Model            | mIoU  ↑         |
> |------|------------------|-----------------|------|------------------|-----------------|
> | HE   | Train-on-TD1      | 5.378 (±0.103)  | LC   | Train-on-OEM      | 68.34 (±0.26)  |
> | HE   | Source only       | 6.117 (±0.385)  | LC   | Source only       | 40.70 (±1.90)  |
> | HE   | DAFormer          | -               | LC   | DAFormer          | 46.29 (±1.18)  |
> | HE   | RS3DAda           | 4.866 (±0.120)  | LC   | RS3DAda           | 48.23 (±1.14)  |
>
> Given the extensive experiments across multiple datasets and tasks in our work, many using ViT-L as the backbone, we honestly did not have the time and resources to run every experiment multiple times during rebuttal phase. However, we will make an effort to include error bars in the revision manuscript where possible.
>
>
> > **Q4:** what does 0 meter GSD mean? According to Table 1 it should actually be 0.09 meters.
>
> **A4:** Thank you for pointing this out. This was an oversight on our part. As you correctly noted, the GSD should be 0.09-1 meters. We appreciate your attention to detail.
>
> > **Q5:** Metrics: I don't know if I've seen sufficient evidence to demonstrate that the proposed \(F_{1}^{HE}\) metric is better than existing metrics. This feels like one-too-many contributions for one paper. This metric has the potential for use for a wide range of regression tasks with long-tail distributions, not just height estimation. This metric deserves its own paper before it's really ready for widespread use
>
> **A5:** Thank you for your insightful suggestion. We fully agree with your perspective. Currently, the metric has only been tested on height estimation, and Figure 1 in the Appendix shows that it aligns more closely with human judgment for this task. Also, as you mentioned, it indeed has the potential to be applied to a broader range of regression tasks with long-tail distributions, though more in-depth work is needed to validate it fully. We will pursue further research on this topic.
>
> > **Q6:** is it the largest synthetic RS dataset (Introduction) or the largest synthetic 3D RS dataset (Abstract)?
>
> **A6:** Thank you for pointing out this. Upon review, we can confirm that our dataset is indeed the largest synthetic remote sensing dataset overall, as stated in the Introduction.
> The confusion stems from an overly specific characterization in the Abstract, which referred to it as the "largest synthetic 3D RS dataset." While this statement is also correct, it inadvertently understates the full scope of our dataset's significance. To avoid any ambiguity, we will update the manuscript to consistently reflect that it is the largest synthetic remote sensing dataset overall.

---

> > ### Author Rebuttal · Authors · 2024-08-17
> >
> > > **Q7:** Abstract: the phrase "easily accessible" is unclear to me. Do you mean that synthetic data is easy to acquire/generate compared to conventional data?
> >
> > **A7:** Thank you for your insightful comment on our terminology. Upon reflection, we concur that "easily accessible" may not fully capture the nuanced advantages of synthetic data in this context. A more precise characterization would be "unrestricted and efficiently annotatable."
> > This rephrasing more accurately reflects our intended meaning:
> >
> > 1) "Unrestricted" encompasses the freedom from privacy, commercial, and licensing constraints,
> > 2) "Efficiently annotatable" highlights the near-effortless nature of the annotation process for synthetic data.
> >
> > The core advantages of synthetic data in our work remain valid, and we believe this more accurate terminology strengthens our argument. We will update the manuscript to clarify this point.
> >
> > > **Q8:** Almost too many figures/tables, feels like SynRS3D and RS3DAda deserve two separate papers.
> > >
> > **A8:** Thank you for your thoughtful consideration of the paper's structure. The decision to combine SynRS3D dataset and RS3DAda method in a single paper was made after careful deliberation. This is motivated by observing that papers on synthetic datasets often struggle to make a significant impact without accompanying methods, which can limit their influence and the potential to inspire follow-up work. Therefore, we combined SynRS3D and RS3DAda into a single paper to increase the usability and impact of SynRS3D. We understand that this may result in content-rich paper. To address this, we will also work on further organizing the figures and tables to improve the readability of the manuscript.
> >
> > > **Q9:** Section 5.1: It wasn't clear to me what the "source-only scenario" was at first until I reread previous sections of the paper. May be worth clarifying the experimental setup again here. I believe it's just "zero shot" prediction, but could be wrong
> >
> > **A9:** Thank you for your attention to detail regarding the terminology used in Section 5.1. We would clarify this term is commonly used in UDA research [1-3] to refer to a model trained solely on the source domain and directly applied to the target domain without adaptation.
> >
> > To avoid confusion, we will revise Section 5.1 to clearly define the "source-only scenario" when it's first mentioned. This will ensure that readers unfamiliar with UDA terminology can easily understand the experimental setup.
> >
> > The term "zero-shot" is more prevalent in foundation model literature. In the context of our work, "source-only" more accurately describes our experimental setup, as it emphasizes the domain shift aspect central to UDA tasks.
> >
> > > **Q10:** The authors document the train-test splits used, but not the exact images in each split. It would be good to either document the random seed used to split the data, or keep a list of images used in each split for reproducibility purposes.
> >
> > **A10:** Thank you for your valuable suggestion on improving reproducibility. We've addressed this by updating our anonymous GitHub repository with cropping scripts for each real-world dataset and lists of datasets used for training and testing, which can be found in `prepare_validation_datasets` directory. The revised manuscript will reference these resources and briefly explain their use.
> >
> > > **Q11:** Why is the batch size so small?
> > >
> > **A11:**  Thank you for your question. We followed the practices of recent advanced UDA algorithms in computer vision, such as DAFormer[1], HRDA[2], and MIC[3], which set their batch sizes to 1 or 2. This approach helps prevent the model from overfitting too quickly to synthetic data and stabilizes the training process, typically paired with a small learning rate. A larger batch size tends to result in a significant drop in accuracy.
> >
> > Additionally, because we use a combination of ViT-L and DPT for the encoder-decoder, even with a batch size of 2, the GPU memory usage reaches 22GB. Therefore, we empirically chose 2 as our batch size.
> >
> > > **Q12:** Corrections regarding figures, tables, formatting, symbols, font sizes, references, and typos in the manuscript.
> >
> > **A12:** We sincerely appreciate the reviewer's meticulous examination of our manuscript and valuable suggestions. Your insights are crucial for enhancing our work's overall quality and clarity. We will revise the manuscript according to your suggestions.
> >
> > **References:**
> >
> > [1] Hoyer, Lukas, Dengxin Dai, and Luc Van Gool. "Daformer: Improving network architectures and training strategies for domain-adaptive semantic segmentation." Proceedings of the IEEE/CVF conference on computer vision and pattern recognition. 2022.
> >
> > [2] Hoyer, Lukas, Dengxin Dai, and Luc Van Gool. "Hrda: Context-aware high-resolution domain-adaptive semantic segmentation." European conference on computer vision. Cham: Springer Nature Switzerland, 2022.
> >
> > [3] Hoyer, Lukas, et al. "MIC: Masked image consistency for context-enhanced domain adaptation." Proceedings of the IEEE/CVF conference on computer vision and pattern recognition. 2023.

---

> > > ### Comment · Reviewer_ezGZ · 2024-08-27
> > > **Increased score and confidence**
> > >
> > > Thank you for one of the best rebuttals I've seen in a long time. You have not only answered all of my questions, but also updated the paper and released additional source code to improve the reproducibility of the paper. I have no further questions or concerns. I have increased both my reviewer score (7 -> 8) and confidence (3 -> 4) now that I better understand the paper.

---

> > > > ### Author Response · Authors · 2024-08-27
> > > >
> > > > Thank you for your exceptionally kind and thoughtful review! We are truly grateful for the time and meticulous attention you have given to our work. Your detailed comments and suggestions have been invaluable in helping us improve the paper and its reproducibility. We are pleased that our rebuttal effectively addressed your concerns and led to an improved evaluation of our work. If you have any further questions, please do not hesitate to reach out. We are always open to feedback and eager to engage in further discussions.

---

### Author Rebuttal · Authors · 2024-08-17

### **Global Response:**

We sincerely appreciate the valuable comments and suggestions from the reviewers. We are grateful for their recognition of the **novelty and significance** of our proposed dataset (R#ezGZ, R#7f97, R#jAqv), the **innovative nature** of our method (R#ezGZ, R#7f97, R#jAqv), the **thoroughness of our experiments and documentation** (R#ezGZ, R#7f97, R#jAqv), the **potential for application** to multiple downstream tasks (R#ezGZ, R#7f97), and the **clarity of our writing** (R#jAqv). We appreciate the reviewers’ careful consideration and positive feedback on these key aspects of our work.

We have carefully addressed the concerns raised by the reviewers and outline our responses and planned improvements below.

---

### **Documentation:**

To address the concerns raised by **reviewers R#jAqv and R#ezGZ** regarding documentation, we **kindly ask** those reviewers who need to reassess the reproducibility of our work to **revisit our anonymous GitHub link.**

> **🔗 These links can be found in lines 5-7 of the Appendix.**

#### **Updates:**

1. **SynRS3D Generation System:** We have decided to open-source the code for our dataset generation system. Detailed instructions can be found in the `synthetic_data_generation` directory.

2. **Evaluation Datasets Preprocessing Code:** We have provided preprocessing code for all the real-world datasets used in our evaluations, along with the images' lists of datasets used in our experiments. Detailed instructions are available in the `prepare_validation_datasets` directory.

Our documentation now comprehensively covers all aspects of our work, including:

1) The complete RS3DAda method code (already included in the initial submission),
2) The SynRS3D dataset link (already included in the initial submission),
3) The newly added SynRS3D generation system code,
4) Newly added preprocessing code for real-world validation datasets.

All these components are available in our anonymous Git repository at **lines 5-7 of the Appendix**. We are committed to open-sourcing these materials to maximize the impact of our work within the community and encourage the reviewers to examine these new resources.

---

### **Relationship Between SynRS3D and RS3DAda:**

SynRS3D and RS3DAda represent complementary advances in remote sensing, addressing the challenge of leveraging synthetic data for real-world applications from distinct yet synergistic angles:

- **SynRS3D** provides an extensive synthetic dataset tailored for remote sensing.
- **RS3DAda** introduces a novel domain adaptation method specifically designed to bridge the synthetic-real data gap in this field.

Together, these contributions form a cohesive solution that significantly enhances the utility of synthetic data in practical remote-sensing scenarios. Our work demonstrates the real-world applicability of SynRS3D, with RS3DAda serving as a compelling proof-of-concept. This synergy not only advances the state-of-the-art but also expands the research landscape at the intersection of synthetic data generation and domain adaptation, facilitating future innovations in remote sensing.

---

### **Scalability and Broader Applications:**

#### **Scalability:**

SynRS3D has been meticulously designed to encompass six distinct city styles from around the world. When combined with RS3DAda, its scalability has been rigorously validated, particularly on the **OEM [1] dataset**—the most diverse feature coverage dataset in the remote sensing domain. OEM spans 77 regions across 44 countries on six continents, providing extensive coverage for land-cover mapping task. Additionally, SynRS3D has been tested on **11 non-synthetic datasets** for height estimation, further demonstrating its robustness.

#### **Broader Applications:**

We would like to draw the attention of **reviewers R#jAqv and R#7f97** to our **Appendix Section A.8 and A.9**, where we explore additional applications beyond land cover mapping and height estimation, such as **building change detection** and **disaster mapping**. During the rebuttal phase, we further demonstrated the breadth of our method and dataset by evaluating **high-resolution canopy height mapping**. We were pleased to find that our results were comparable to those from study [2], which trained a foundation model using a vast amount of real-world data. This further proves the broad applicability of our dataset and method.

---

### **Newly Uploaded One-Page PDF:**

We have included photorealistic visualizations of SynRS3D after style transfer (Figure 1) and results in high-resolution canopy height mapping (Figure 2) in the attached PDF.

---

**Real and open-source high-resolution data** is inherently difficult to access in remote sensing, as it is often restricted by data usage protocols from commercial companies. Given these unique challenges, synthetic data like SynRS3D is invaluable. Our dataset has demonstrated impressive performance across various benchmarks, underscoring its potential for real-world applications.

---

We once again sincerely thank all the reviewers for their constructive comments, generous recognition, and the effort they put into reviewing our manuscript.

**References:**

[1] Xia, Junshi, et al. "Openearthmap: A benchmark dataset for global high-resolution land cover mapping." Proceedings of the IEEE/CVF Winter Conference on Applications of Computer Vision. 2023.

[2] Tolan, Jamie, et al. "Very high resolution canopy height maps from RGB imagery using self-supervised vision transformer and convolutional decoder trained on aerial lidar." Remote Sensing of Environment 300 (2024): 113888.

---

### Decision · Program_Chairs · 2024-09-26

**Decision:**

Accept (Spotlight)

**Comment:**

Dear Authors,

I have reviewed the comments from the reviewers and your response.

I am pleased to recommend the acceptance of this paper. Overall, your work and also the detailed rebuttal are highly appreciated by the reviewers. To cite one of the reviewers: “Thank you for one of the best rebuttals I've seen in a long time.” This is impressive!

Specifically, multiple reviewers praise the introduction of the SynRS3D dataset, the largest synthetic dataset for 3D remote sensing, as a significant contribution to Earth Observation. The introduction of the RS3DAda domain adaptation method is noted as a novel contribution. The experiments and documentation are very thorough.

To further close the gap between synthetic data and real-world data, reviewers suggested to use style transfer. Also, expanding validation across more diverse real-world datasets and regions is a common suggestion.

I hope these comments can be helpful to further improve your great work.